# Model Evolution Under Zeroth-Order Optimization: A Neural Tangent Kernel Perspective

**Chen Zhang**[1]* **Yuxin Cheng**[1]* **Chenchen Ding**[1] **Shuqi Wang**[1] **Jingreng Lei**[1]
**Runsheng Yu**[2] **Yik-Chung Wu**[1] **Ngai Wong**[1]

[1] The University of Hong Kong
[2] The Hong Kong University of Science and Technology

## Abstract

Zeroth-order (ZO) optimization enables memory-efficient training of neural networks by estimating gradients via forward passes only, eliminating the need for backpropagation. However, the stochastic nature of gradient estimation significantly obscures the training dynamics, in contrast to the well-characterized behavior of first-order methods under Neural Tangent Kernel (NTK) theory. To address this, we introduce the Neural Zeroth-order Kernel (NZK) to describe model evolution in function space under ZO updates. For linear models, we prove that the expected NZK remains constant throughout training and depends explicitly on the first and second moments of the random perturbation directions. This invariance yields a closed-form expression for model evolution under squared loss. We further extend the analysis to linearized neural networks. Interpreting ZO updates as kernel gradient descent via NZK provides a novel perspective for potentially accelerating convergence. Extensive experiments across synthetic and real-world datasets (including MNIST, CIFAR-10, and Tiny ImageNet) validate our theoretical results and demonstrate acceleration when using a single shared random vector. Codes are available at the LINK.

## 1 Introduction and Related Work

Zeroth-order (ZO) optimization approximates gradients using finite differences computed via forward passes, offering a memory-efficient alternative to first-order (FO) methods and enabling optimization of non-differentiable objectives (Nesterov & Spokoiny, 2017; Duchi et al., 2015; Liu et al., 2020b; Malladi et al., 2023; Wang et al., 2025). Despite strong empirical performance in black-box attacks, LLM fine-tuning, and on-chip learning, theoretical understanding of ZO training dynamics remains limited due to inherent randomness.

In contrast, FO methods benefit from Neural Tangent Kernel (NTK) theory (Jacot et al., 2018; Lee et al., 2019), which shows that wide neural networks evolve linearly in function space with a constant NTK, yielding closed-form dynamics under squared loss. Recent works have further bridged NTK theory with nonparametric teaching frameworks (Zhang et al., 2023b;a) to accelerate training dynamics in various neural architectures (Zhang et al., 2024; 2025; 2026a;b). NTK theory, however, relies on exact gradients and cannot be directly applied to ZO.

To bridge this gap, we propose the Neural Zeroth-order Kernel (NZK), which adapts the NTK concept to describe model evolution in function space under random directional perturbations. Starting from linear models, we reveal that (i) the expected NZK is time-invariant, (ii) its form depends explicitly on the moments of random directions, and (iii) reusing the same random vector for perturbation and Jacobian estimation yields substantial convergence acceleration. We then extend these insights to linearized neural networks, providing a foundation for understanding ZO dynamics in wider regimes. Our key contributions are: **(1)** Introduction of the Neural Zeroth-order Kernel (NZK) for analyzing ZO optimization in function space; **(2)** Closed-form evolution dynamics for linear and linearized

---

*Equal contribution

models under squared loss; **(3)** We empirically validate our theoretical findings through extensive experiments (MNIST, CIFAR-10, Tiny ImageNet).

## 2 PRELIMINARY

We consider empirical risk minimization for a scalar-valued model $f(\boldsymbol{x}; \theta)$ with parameters $\theta \in \mathbb{R}^d$. ZO gradient descent updates parameters as

$$\theta_{t+1} \leftarrow \theta_t - \eta \mathcal{G}_t, \quad \mathcal{G}_t = \frac{1}{N} \sum_{i=1}^N \frac{\mathcal{L}(f(\boldsymbol{x}_i; \theta_t + \epsilon \boldsymbol{z})) - \mathcal{L}(f(\boldsymbol{x}_i; \theta_t - \epsilon \boldsymbol{z}))}{2\epsilon} \boldsymbol{z}, \tag{1}$$

where $\boldsymbol{z} \sim \mathcal{N}(\boldsymbol{0}, \sigma^2 \boldsymbol{I}_d)$ (unless stated otherwise) and $\epsilon > 0$ is small.

The Neural Tangent Kernel (NTK) (Jacot et al., 2018) describes FO dynamics in function space via $K(\boldsymbol{x}_i, \boldsymbol{x}_j) = \left\langle \frac{\partial f}{\partial \theta} \big|_{\boldsymbol{x}_i}, \frac{\partial f}{\partial \theta} \big|_{\boldsymbol{x}_j} \right\rangle$, which remains constant in wide networks, enabling closed-form analysis.

## 3 NEURAL ZEROTH-ORDER KERNEL

### 3.1 EVOLUTION OF LINEAR MODELS UNDER ZO OPTIMIZATION

Our primary focus lies on the evolution of the linear model $f(\boldsymbol{x}; \theta) = \langle \theta, \boldsymbol{x} \rangle$ itself rather than the parameter dynamics. It is important to recognize that gradients in FO methods can be decomposed into two components: a factor governing magnitude and a vector dictating direction (Ruder, 2016; Zhang et al., 2023b;a; 2024), which may not necessarily be unitary. Building on this understanding, we reformulate $\mathcal{G}_t$ in Eq. 1 equivalently as:

$$\mathcal{G}_t = \frac{1}{N} \underbrace{\left[ \frac{\mathcal{L}(f(\boldsymbol{x}_i; \theta_t + \epsilon \boldsymbol{z}), y_i) - \mathcal{L}(f(\boldsymbol{x}_i; \theta_t - \epsilon \boldsymbol{z}), y_i)}{f(\boldsymbol{x}_i; \theta_t + \epsilon \boldsymbol{z}) - f(\boldsymbol{x}_i; \theta_t - \epsilon \boldsymbol{z})} \right]_N^\top}_{(*)} \cdot \underbrace{\left[ \frac{f(\boldsymbol{x}_i; \theta_t + \epsilon \boldsymbol{z}) - f(\boldsymbol{x}_i; \theta_t - \epsilon \boldsymbol{z})}{2\epsilon} \boldsymbol{z} \right]_N}_{(**)}. \tag{2}$$

The magnitude relates to $(*)$, whereas the direction corresponds to $(**)$. $(*)$ allows for $\mathcal{L}$ to potentially be non-differentiable (Stiennon et al., 2020; Ouyang et al., 2022), contrasting with FO methods. Meanwhile, one can see that training involves the entire dataset but uses only one $\boldsymbol{z}$ per iteration, which means the gradient is estimated across numerous data points using a single random direction $\boldsymbol{z}$.

Meanwhile, the rate of change of $f$ w.r.t. $\theta$, denoted as $\partial f_\theta / \partial \theta$ can be expressed through finite differences in ZO optimization as:

$$\frac{f(\cdot; \theta_t + \epsilon \boldsymbol{\zeta}) - f(\cdot; \theta_t - \epsilon \boldsymbol{\zeta})}{2\epsilon} \boldsymbol{\zeta}, \tag{3}$$

where $\boldsymbol{\zeta}$ signifies the same concept as $\boldsymbol{z}$. Therefore, the evolution of $f(\boldsymbol{x}; \theta)$ can be delineated as:

$$f_{\theta_{t+1}} - f_{\theta_t} = -\frac{\eta}{N} \left[ \frac{\mathcal{L}(f(\boldsymbol{x}_i; \theta_t + \epsilon \boldsymbol{z}), y_i) - \mathcal{L}(f(\boldsymbol{x}_i; \theta_t - \epsilon \boldsymbol{z}), y_i)}{f(\boldsymbol{x}_i; \theta_t + \epsilon \boldsymbol{z}) - f(\boldsymbol{x}_i; \theta_t - \epsilon \boldsymbol{z})} \right]_N^\top \cdot [K_{\boldsymbol{\zeta}, \boldsymbol{z}}(\cdot, \boldsymbol{x}_i)]_N, \tag{4}$$

where $[K_{\boldsymbol{\zeta}, \boldsymbol{z}}(\cdot, \boldsymbol{x}_i)]_N$ represents a column vector extracted from a random kernel matrix $\boldsymbol{K}_{\boldsymbol{\zeta}, \boldsymbol{z}}$. This kernel matrix $\boldsymbol{K}_{\boldsymbol{\zeta}, \boldsymbol{z}}$ is referred to as the Neural Zeroth-order Kernel (NZK). Its entries are

$$K_{\boldsymbol{\zeta}, \boldsymbol{z}}(\boldsymbol{x}_i, \boldsymbol{x}_j) = \left\langle \frac{f(\boldsymbol{x}_i; \theta_t + \epsilon \boldsymbol{\zeta}) - f(\boldsymbol{x}_i; \theta_t - \epsilon \boldsymbol{\zeta})}{2\epsilon} \boldsymbol{\zeta}, \frac{f(\boldsymbol{x}_j; \theta_t + \epsilon \boldsymbol{z}) - f(\boldsymbol{x}_j; \theta_t - \epsilon \boldsymbol{z})}{2\epsilon} \boldsymbol{z} \right\rangle. \tag{5}$$

The stability of the NTK is crucial for analyzing how neural networks evolve using FO methods, an area that has seen significant research efforts (Jacot et al., 2018; Liu et al., 2020a). We show the stability of the expected NZK, and we also establish the connection between the expected NZK and the distribution of random direction vectors.

**Theorem 1.** *Suppose a linear model $f(\boldsymbol{x}; \theta)$ is trained using ZO optimization with random direction vectors $\boldsymbol{z} \sim \mathcal{N}(\mu_{\boldsymbol{z}} \boldsymbol{1}, \sigma_{\boldsymbol{z}}^2 \boldsymbol{I}_d)$, and the rate of change of $f(\boldsymbol{x}; \theta)$ w.r.t. $\theta$ is estimated following Eq. 3 with $\boldsymbol{\zeta} \sim \mathcal{N}(\mu_{\boldsymbol{\zeta}} \boldsymbol{1}, \sigma_{\boldsymbol{\zeta}}^2 \boldsymbol{I}_d)$, the corresponding NZK stays constant throughout training in its expected sense, with its entries closely tied to the distribution of the random direction vectors.*

$$\mathbb{E}_{\boldsymbol{\zeta}, \boldsymbol{z}} K_{\boldsymbol{\zeta}, \boldsymbol{z}}(\boldsymbol{x}_i, \boldsymbol{x}_j) = \sigma_{\boldsymbol{\zeta}}^2 \sigma_{\boldsymbol{z}}^2 \langle \boldsymbol{x}_i, \boldsymbol{x}_j \rangle + \sigma_{\boldsymbol{\zeta}}^2 \mu_{\boldsymbol{z}}^2 \langle \boldsymbol{x}_i, \boldsymbol{1} \rangle \langle \boldsymbol{1}, \boldsymbol{x}_j \rangle + \mu_{\boldsymbol{\zeta}}^2 \sigma_{\boldsymbol{z}}^2 \langle \boldsymbol{x}_i, \boldsymbol{1} \rangle \langle \boldsymbol{1}, \boldsymbol{x}_j \rangle + d \mu_{\boldsymbol{\zeta}}^2 \mu_{\boldsymbol{z}}^2 \langle \boldsymbol{x}_i, \boldsymbol{1} \rangle \langle \boldsymbol{1}, \boldsymbol{x}_j \rangle. \tag{6}$$

The purpose of using the expectation operation here is twofold: to control the randomness in estimation and to unveil a statistical characteristic of the NZK. Meanwhile, this can be achieved by utilizing batch sampling (Duchi et al., 2015; Liu et al., 2020c), with a sufficiently large batch size, as per the law of large numbers. To simplify the notation, we define symmetric and positive definite $\mathcal{K}_{\boldsymbol{\zeta},\boldsymbol{z}} := \mathbb{E}_{\boldsymbol{\zeta},\boldsymbol{z}} \boldsymbol{K}_{\boldsymbol{\zeta},\boldsymbol{z}}$.

Typically, in most literature (see Liu et al., 2020b for a survey), the components of $\boldsymbol{z}$ and $\boldsymbol{\zeta}$ are assumed to have zero mean and unit variance, *i.e.*, $\mu_{\boldsymbol{z}} = \mu_{\boldsymbol{\zeta}} = 0$ and $\sigma_{\boldsymbol{z}}^2 = \sigma_{\boldsymbol{\zeta}}^2 = 1$. Under these conditions, Eq. 6 simplifies to $\mathcal{K}_{\boldsymbol{\zeta},\boldsymbol{z}}(\boldsymbol{x}_i, \boldsymbol{x}_j) = \langle \boldsymbol{x}_i, \boldsymbol{x}_j \rangle$, implying that the expected NZK equals its FO counterpart, *i.e.*, NTK (Jacot et al., 2018; Zhang et al., 2024; 2025; 2026a;b). We highlight that this kernel equivalence pertains to the function space. Interestingly, this aligns with the unbiasedness of gradient estimation derived in parameter space (Nesterov & Spokoiny, 2017; Berahas et al., 2022).

Since $\mathcal{K}_{\boldsymbol{\zeta},\boldsymbol{z}}$ does not vary with time, when the loss function is specified as the squared loss, the model dynamics following Eq. 4 can be expressed in closed form as:

$$[f_{\theta_t}(\boldsymbol{x}_i)]_N = \left( \boldsymbol{I}_N - \left( \boldsymbol{I}_N - \eta\bar{\mathcal{K}}_{\boldsymbol{\zeta},\boldsymbol{z}} \right)^t \right) [f^*(\boldsymbol{x}_i)]_N + \left( \boldsymbol{I}_N - \eta\bar{\mathcal{K}}_{\boldsymbol{\zeta},\boldsymbol{z}} \right)^t [f_{\theta_0}(\boldsymbol{x}_i)]_N, \tag{7}$$

where $\bar{\mathcal{K}}_{\boldsymbol{\zeta},\boldsymbol{z}} = \mathcal{K}_{\boldsymbol{\zeta},\boldsymbol{z}}/N$. The detailed derivation is deferred to Appendix A.2. This closed form remains consistent with that of FO methods, with the modification of replacing $\bar{\mathcal{K}}_{\boldsymbol{\zeta},\boldsymbol{z}}$ by $\bar{\boldsymbol{K}}$. This suggests that, in terms of expected behavior, linear models evolve similarly under both ZO and FO optimization.

Interestingly, although $\boldsymbol{\zeta}$ and $\boldsymbol{z}$ are distinct independent random vectors serving different purposes, they originate from the same $d$-dimensional space. This naturally allows to generate either $\boldsymbol{\zeta}$ or $\boldsymbol{z}$ and apply it to the other. This approach saves memory and results in an accelerated performance.

**Corollary 2.** *When $\boldsymbol{\zeta}$ and $\boldsymbol{z}$ are identical random vectors with zero mean and independent entries $z_i, i \in \mathbb{N}_d$, meaning using a single random vector for ZO optimization and estimating the rate of change of $f(\boldsymbol{x};\theta)$ w.r.t. $\theta$, we have*

$$\mathcal{K}_{\boldsymbol{\zeta},\boldsymbol{z}}(\boldsymbol{x}_i, \boldsymbol{x}_j) = \left( \mathbb{V}z_i^2 + d \cdot \mathbb{E}^2 \left[ z_i^2 \right] \right) \langle \boldsymbol{x}_i, \boldsymbol{x}_j \rangle, \tag{8}$$

*where $\mathbb{V}$ denotes the variance operation. If $\boldsymbol{\zeta} = \boldsymbol{z} \sim \mathcal{N}(\boldsymbol{0}, \sigma_{\boldsymbol{z}}^2 \boldsymbol{I}_d)$, we have*

$$\mathcal{K}_{\boldsymbol{\zeta},\boldsymbol{z}}(\boldsymbol{x}_i, \boldsymbol{x}_j) = (d+2)\sigma_{\boldsymbol{z}}^4 \langle \boldsymbol{x}_i, \boldsymbol{x}_j \rangle, \tag{9}$$

*and $(z_i/\sigma_{\boldsymbol{z}})^2 \sim \chi^2(1)$.*

It can be observed that by setting $\boldsymbol{z} = \boldsymbol{\zeta} \sim \mathcal{N}(\boldsymbol{0}, \sigma_{\boldsymbol{z}}^2 \boldsymbol{I}_d)$, the expected NZK is scaled by $(d+2)\sigma_{\boldsymbol{z}}^4$. This implies the potential to accelerate ZO optimization.

## 3.2 Extension to linearized neural networks

Following (Lee et al., 2019), we consider linearized networks (without specifying the argument $\boldsymbol{x}$)

$$f^{\text{lin}}(\theta_t) \equiv f(\theta_0) + \left\langle \frac{f(\theta_0 + \epsilon\boldsymbol{u}) - f(\theta_0 - \epsilon\boldsymbol{u})}{2\epsilon} \boldsymbol{u}, \theta_t - \theta_0 \right\rangle, \tag{10}$$

To characterize the training dynamics of $f^{\text{lin}}(\boldsymbol{x};\theta_t)$ within the function space, it is necessary to derive the corresponding expected NZK, which is

$$\mathcal{K}_{\boldsymbol{u},\boldsymbol{\zeta},\boldsymbol{z}}(\boldsymbol{x}_i, \boldsymbol{x}_j) = \left\langle \widehat{\frac{\partial f(\boldsymbol{x}_i;\theta)}{\partial \theta_0}}, \widehat{\frac{\partial f(\boldsymbol{x}_j;\theta)}{\partial \theta_0}} \right\rangle, \tag{11}$$

where the random direction vectors are assumed to have zero mean and unit variance, and

$$\widehat{\frac{\partial f(\boldsymbol{x}_i;\theta)}{\partial \theta_0}} := \mathbb{E}_{\boldsymbol{u}} \frac{f(\boldsymbol{x}_i;\theta_0 + \epsilon\boldsymbol{u}) - f(\boldsymbol{x}_i;\theta_0 - \epsilon\boldsymbol{u})}{2\epsilon} \boldsymbol{u}.$$

The term $\mathcal{K}_{\boldsymbol{u},\boldsymbol{\zeta},\boldsymbol{z}}(\boldsymbol{x}_i, \boldsymbol{x}_j)$ differs from, yet shares similarities with, the expected NZK of linear models, in the sense that it replaces the input $\boldsymbol{x}$ with the estimation of the FO neural tangent random feature.

It is clear that this expected NZK remains unchanged during training. Consequently, for squared loss, we can derive the closed-form evolution of linearized neural networks as:

$$\left[ f_{\theta_t}^{\text{lin}}(\boldsymbol{x}_i) \right]_N = \left( \boldsymbol{I}_N - \left( \boldsymbol{I}_N - \eta\bar{\mathcal{K}}_{\boldsymbol{u},\boldsymbol{\zeta},\boldsymbol{z}} \right)^t \right) [f^*(\boldsymbol{x}_i)]_N + \left( \boldsymbol{I}_N - \eta\bar{\mathcal{K}}_{\boldsymbol{u},\boldsymbol{\zeta},\boldsymbol{z}} \right)^t [f_{\theta_0}(\boldsymbol{x}_i)]_N. \tag{12}$$

## 4 EXPERIMENTS

We validate our findings in student-teacher settings and on both synthetic and real datasets. Additional discussion is provided in Appendix C.

**Linear models**. In practice, $\mu_z$ is commonly set to zero (Liu et al., 2020b), a setting we adopt in our experiments. We consider a linear model $f(\boldsymbol{x}) = \langle \theta, \boldsymbol{x} \rangle$ with $\theta \in \mathbb{R}^2$. The teacher model is defined as $f^*(\boldsymbol{x}; \theta^*) = 7.0 * x_1 + 2.0 * x_2 + \delta$, where $\delta$ represents Gaussian perturbation. The training set consists of data points $\boldsymbol{x}$ randomly sampled from a unit circle, *i.e.*, $\boldsymbol{x} \in \mathbb{S}^2$. The parameter $\theta$ of the student model $f_0(\boldsymbol{x}; \theta)$ is initialized randomly from a Gaussian

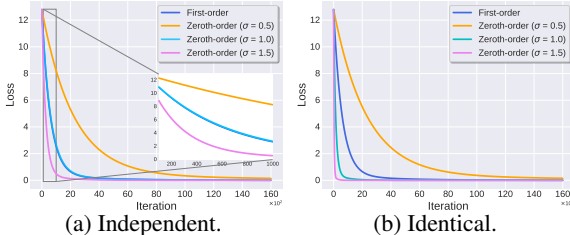

(a) Independent.  (b) Identical.

Figure 1: Comparison of losses in 2-D tasks under FO and ZO with varying $\sigma_z$ for sampling. "Independent" indicates that $\boldsymbol{z}$ and $\boldsymbol{\zeta}$ are sampled independently, while "Identical" denotes that $\boldsymbol{\zeta}$ remains the same as $\boldsymbol{z}$.

distribution $\mathcal{N}(\boldsymbol{0}, \boldsymbol{I}_2)$. Throughout training, we maintain a fixed learning rate $\eta = $1e-3 and iterations for both FO and ZO optimization. For ZO optimization, we set $\epsilon = $1e-3, sample $\boldsymbol{\zeta}$ from $\mathcal{N}(\boldsymbol{0}, \boldsymbol{I}_2)$, and estimate $\mathcal{K}_{\boldsymbol{\zeta}, \boldsymbol{z}}(\boldsymbol{x}_i, \boldsymbol{x}_j)$ by averaging over 10,000 random samples.

Figure 1 (a) shows that with $\sigma_z = 1$, both FO and ZO exhibit similar convergence rates. Besides, we find that for ZO, the evolution rate accelerates as $\sigma_z$ increases. When considering the scenario where $\boldsymbol{\zeta}$ and $\boldsymbol{z}$ are identical, using a single random vector for zeroth-order optimization and estimating the rate of change of $f(\boldsymbol{x}; \theta)$ w.r.t. $\theta$, Figure 1 (b) shows that this memory-saving method can also speed up convergence without necessitating an increase in variance. The visualization of model evolutions can be found in Figure 5. These align with Theorem 1 and Corollary 2.

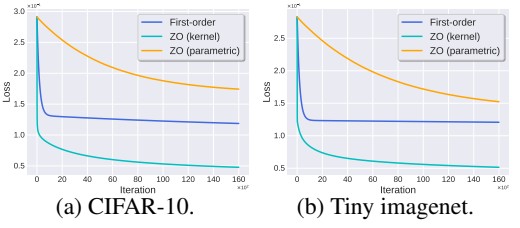
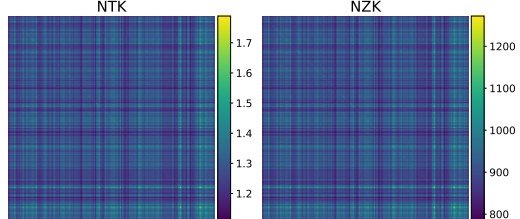

(a) CIFAR-10.  (b) Tiny imagenet.

Figure 2: Comparison of losses on CIFAR-10 and Tiny imagenet using the linearized neural network under FO, traditional parametric gradient ZO ("ZO (parametric)") and kernel gradient ZO with identical $\boldsymbol{\zeta}$ and $\boldsymbol{z}$ ("ZO (kernel)").

Figure 3: Comparison of NZK for linearized neural networks in tiny imagenet classification using FO (left) and ZO (right), with identical $\boldsymbol{\zeta}$ and $\boldsymbol{z}$.

**Linearized neural networks**. We also perform practical classification tasks on the CIFAR-10 and tiny imagenet dataset using the linearized neural network. The outcomes of these tasks highlight NZK's capability to handle more complex tasks and emphasize the effectiveness of examining ZO using kernel gradients. The loss plots shown in Figure 2 tell that updating with traditional parametric gradient ZO converges more slowly than FO, which in turn performs worse than using kernel gradients ZO. This finding is consistent with our theoretical results. The visualization of NZK on tiny imagenet can be found in Figure 3, while that for CIFAR-10 are presented in Figure 18.

## 5 CONCLUSION

We introduced the Neural Zeroth-order Kernel to analyze ZO optimization in function space. The expected NZK is invariant and moment-dependent, yielding closed-form dynamics and suggesting a novel acceleration strategy via shared random directions. These findings pave the way for linking ZO dynamics to recent research analyzing neural networks from a functional perspective (Zhang et al., 2024; 2025; 2026a;b).

## ACKNOWLEDGMENTS

This work was supported in part by the Theme-based Research Scheme (TRS) project T45-701/22-R of the Research Grants Council of Hong Kong, and in part by the AVNET-HKU Emerging Microelectronics and Ubiquitous Systems (EMUS) Lab.

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

# Appendix

## A  ADDITIONAL DISCUSSIONS

### A.1  NEURAL ZEROTH-ORDER KERNEL (NZK)

It follows from 1 that we can formulate the parameter dynamics as

$$
\begin{aligned}
\theta_{t+1} - \theta_t &= -\frac{\eta}{N} \sum_{i=1}^{N} \frac{\mathcal{L}(f(\boldsymbol{x}_i; \theta_t + \epsilon \boldsymbol{z}), y_i) - \mathcal{L}(f(\boldsymbol{x}_i; \theta_t - \epsilon \boldsymbol{z}), y_i)}{2\epsilon} \cdot \boldsymbol{z} \\
&= -\frac{\eta}{N} \underbrace{\left[ \frac{\mathcal{L}(f(\boldsymbol{x}_i; \theta_t + \epsilon \boldsymbol{z}), y_i) - \mathcal{L}(f(\boldsymbol{x}_i, \theta_t - \epsilon \boldsymbol{z}), y_i)}{f(\boldsymbol{x}_i; \theta_t + \epsilon \boldsymbol{z}) - f(\boldsymbol{x}_i; \theta_t - \epsilon \boldsymbol{z})} \right]_N^T}_{(*)} \underbrace{\left[ \frac{f(\boldsymbol{x}_i; \theta_t + \epsilon \boldsymbol{z}) - f(\boldsymbol{x}_i; \theta_t - \epsilon \boldsymbol{z})}{2\epsilon} \cdot \boldsymbol{z} \right]_N}_{(**)} \quad (13)
\end{aligned}
$$

where $(*)$ stands for estimating the linear approximation of $\left. \frac{\partial \mathcal{L}}{\partial f_\theta} \right|_{f_{\theta_t}, \boldsymbol{x}_i}$, influencing the magnitude in FO gradients, while $(**)$ relates to the direction vector.

Using Equation 3, the estimation of how $f_\theta$ changes concerning $\theta$, and applying the chain rule, we can describe the evolution of the model $f_\theta$ as

$$
f_{\theta_{t+1}} - f_{\theta_t} = \left\langle \frac{f(\cdot; \theta_t + \epsilon \boldsymbol{\zeta}) - f(\cdot; \theta_t - \epsilon \boldsymbol{\zeta})}{2\epsilon} \boldsymbol{\zeta}, \theta_{t+1} - \theta_t \right\rangle. \quad (14)
$$

By substituting Equation 13 into the preceding equation, we obtain

$$
\begin{aligned}
f_{\theta_{t+1}} - f_{\theta_t} &= -\frac{\eta}{N} \left\langle \frac{f(\cdot; \theta_t + \epsilon \boldsymbol{\zeta}) - f(\cdot; \theta_t - \epsilon \boldsymbol{\zeta})}{2\epsilon} \boldsymbol{\zeta}, \right. \\
&\qquad \left. \left[ \frac{\mathcal{L}(f(\boldsymbol{x}_i; \theta_t + \epsilon \boldsymbol{z}), y_i) - \mathcal{L}(f(\boldsymbol{x}_i; \theta_t - \epsilon \boldsymbol{z}), y_i)}{f(\boldsymbol{x}_i; \theta_t + \epsilon \boldsymbol{z}) - f(\boldsymbol{x}_i; \theta_t - \epsilon \boldsymbol{z})} \right]_N^T \left[ \frac{f(\boldsymbol{x}_i; \theta_t + \epsilon \boldsymbol{z}) - f(\boldsymbol{x}_i; \theta_t - \epsilon \boldsymbol{z})}{2\epsilon} \cdot \boldsymbol{z} \right]_N \right\rangle \\
&= -\frac{\eta}{N} \left[ \frac{\mathcal{L}(f(\boldsymbol{x}_i; \theta_t + \epsilon \boldsymbol{z}), y_i) - \mathcal{L}(f(\boldsymbol{x}_i; \theta_t - \epsilon \boldsymbol{z}), y_i)}{f(\boldsymbol{x}_i; \theta_t + \epsilon \boldsymbol{z}) - f(\boldsymbol{x}_i; \theta_t - \epsilon \boldsymbol{z})} \right]_N^T \\
&\qquad \left\langle \frac{f(\cdot; \theta_t + \epsilon \boldsymbol{\zeta}) - f(\cdot; \theta_t - \epsilon \boldsymbol{\zeta})}{2\epsilon} \boldsymbol{\zeta}, \left[ \frac{f(\boldsymbol{x}_i; \theta_t + \epsilon \boldsymbol{z}) - f(\boldsymbol{x}_i; \theta_t - \epsilon \boldsymbol{z})}{2\epsilon} \cdot \boldsymbol{z} \right]_N \right\rangle \\
&= -\frac{\eta}{N} \left[ \frac{\mathcal{L}(f(\boldsymbol{x}_i; \theta_t + \epsilon \boldsymbol{z}), y_i) - \mathcal{L}(f(\boldsymbol{x}_i; \theta_t - \epsilon \boldsymbol{z}), y_i)}{f(\boldsymbol{x}_i; \theta_t + \epsilon \boldsymbol{z}) - f(\boldsymbol{x}_i; \theta_t - \epsilon \boldsymbol{z})} \right]_N^T \\
&\qquad \left[ \underbrace{\left\langle \frac{f(\cdot; \theta_t + \epsilon \boldsymbol{\zeta}) - f(\cdot; \theta_t - \epsilon \boldsymbol{\zeta})}{2\epsilon} \boldsymbol{\zeta}, \frac{f(\boldsymbol{x}_i; \theta_t + \epsilon \boldsymbol{z}) - f(\boldsymbol{x}_i; \theta_t - \epsilon \boldsymbol{z})}{2\epsilon} \cdot \boldsymbol{z} \right\rangle}_{(\Xi)} \right]_N, \quad (15)
\end{aligned}
$$

where $(\Xi)$ defines a kernel $\boldsymbol{K}_{\boldsymbol{\zeta}, \boldsymbol{z}}$ on the training set. This kernel matrix $\boldsymbol{K}_{\boldsymbol{\zeta}, \boldsymbol{z}}$ is named the Neural Zeroth-order Kernel, and its entries are defined as

$$
K_{\boldsymbol{\zeta}, \boldsymbol{z}}(\boldsymbol{x}_i, \boldsymbol{x}_j) = \left\langle \frac{f(\boldsymbol{x}_i; \theta_t + \epsilon \boldsymbol{\zeta}) - f(\boldsymbol{x}_i; \theta_t - \epsilon \boldsymbol{\zeta})}{2\epsilon} \boldsymbol{\zeta}, \frac{f(\boldsymbol{x}_j; \theta_t + \epsilon \boldsymbol{z}) - f(\boldsymbol{x}_j; \theta_t - \epsilon \boldsymbol{z})}{2\epsilon} \boldsymbol{z} \right\rangle. \quad (16)
$$

In simple terms, the correlation between NZK and NTK can be informally explained via the perturbation $\epsilon$. Specifically, for differentiable models $f(\boldsymbol{x}; \theta)$, NTK is included in the NZK set with respect to random variables $\boldsymbol{\zeta}$ and $\boldsymbol{z}$ when the perturbation becomes infinitesimal (Flaxman et al., 2005), that is,

$$
K(\boldsymbol{x}_i, \boldsymbol{x}_j) \in \lim_{\epsilon \to 0} K_{\boldsymbol{\zeta}, \boldsymbol{z}}(\boldsymbol{x}_i, \boldsymbol{x}_j). \quad (17)
$$

## A.2 Derivation of Equation 7, *i.e.*, the closed form of model dynamics

In the case of squared loss, represented as $\mathcal{L}(f, f^*) = \frac{1}{2}(f - f^*)^2$, and with $f^*(\boldsymbol{x}_i) \equiv y_i$ signifying the target of $\boldsymbol{x}$, Equation 4 can be expressed as

$$
\begin{aligned}
f_{\theta_{t+1}} - f_{\theta_t} &= -\frac{\eta}{2N} \left[ \frac{(f(\boldsymbol{x}_i; \theta_t + \epsilon \boldsymbol{z}) - f^*(\boldsymbol{x}_i))^2 - (f(\boldsymbol{x}_i; \theta_t - \epsilon \boldsymbol{z}) - f^*(\boldsymbol{x}_i))^2}{f(\boldsymbol{x}_i; \theta_t + \epsilon \boldsymbol{z}) - f(\boldsymbol{x}_i; \theta_t - \epsilon \boldsymbol{z})} \right]_N^T \cdot [K_{\boldsymbol{\varsigma},\boldsymbol{z}}(\cdot, \boldsymbol{x}_i)]_N \\
&= -\frac{\eta}{2N} [f(\boldsymbol{x}_i; \theta_t + \epsilon \boldsymbol{z}) + f(\boldsymbol{x}_i; \theta_t - \epsilon \boldsymbol{z}) - 2f^*(\boldsymbol{x}_i)]_N^T \cdot [K_{\boldsymbol{\varsigma},\boldsymbol{z}}(\cdot, \boldsymbol{x}_i)]_N .
\end{aligned} \tag{18}
$$

Regarding the summation term $f(\boldsymbol{x}; \theta_t + \epsilon \boldsymbol{z}) + f(\boldsymbol{x}; \theta_t - \epsilon \boldsymbol{z})$, we obtain

$$
\begin{aligned}
f(\boldsymbol{x}; \theta_t + \epsilon \boldsymbol{z}) + f(\boldsymbol{x}; \theta_t - \epsilon \boldsymbol{z}) &= \langle \theta_t + \epsilon \boldsymbol{z}, \boldsymbol{x} \rangle + \langle \theta_t - \epsilon \boldsymbol{z}, \boldsymbol{x} \rangle \\
&= \langle 2\theta_t, \boldsymbol{x} \rangle \\
&= 2f(\boldsymbol{x}; \theta_t).
\end{aligned} \tag{19}
$$

Hence, we can simplify Equation 18 to

$$
\begin{aligned}
f_{\theta_{t+1}} - f_{\theta_t} &= -\frac{\eta}{N} [f(\boldsymbol{x}_i; \theta_t) - f^*(\boldsymbol{x}_i)]_N^T \cdot [K_{\boldsymbol{\varsigma},\boldsymbol{z}}(\cdot, \boldsymbol{x}_i)]_N \\
&= -\frac{\eta}{N} [f_{\theta_t}(\boldsymbol{x}_i) - f^*(\boldsymbol{x}_i)]_N^T \cdot [K_{\boldsymbol{\varsigma},\boldsymbol{z}}(\cdot, \boldsymbol{x}_i)]_N .
\end{aligned} \tag{20}
$$

After taking the expectation, we define $\mathcal{K}_{\boldsymbol{\varsigma},\boldsymbol{z}}$ as $\mathbb{E}_{\boldsymbol{\varsigma},\boldsymbol{z}} \boldsymbol{K}_{\boldsymbol{\varsigma},\boldsymbol{z}}$. According to Theorem 1, it can be seen that $\mathcal{K}_{\boldsymbol{\varsigma},\boldsymbol{z}}$ remains constant over time. By substituting the inputs into Equation 20, the expected model dynamics can be written as

$$
\begin{aligned}
\left[ f_{\theta_{t+1}}(\boldsymbol{x}_i) - f_{\theta_t}(\boldsymbol{x}_i) \right]_N &= -\frac{\eta}{N} \mathcal{K}_{\boldsymbol{\varsigma},\boldsymbol{z}} \cdot [f_{\theta_t}(\boldsymbol{x}_i) - f^*(\boldsymbol{x}_i)]_N \\
\therefore \quad \left[ f_{\theta_{t+1}}(\boldsymbol{x}_i) - f^*(\boldsymbol{x}_i) \right]_N - [f_{\theta_t}(\boldsymbol{x}_i) - f^*(\boldsymbol{x}_i)]_N &= -\frac{\eta}{N} \mathcal{K}_{\boldsymbol{\varsigma},\boldsymbol{z}} \cdot [f_{\theta_t}(\boldsymbol{x}_i) - f^*(\boldsymbol{x}_i)]_N \\
\therefore \quad \left[ f_{\theta_{t+1}}(\boldsymbol{x}_i) - f^*(\boldsymbol{x}_i) \right]_N &= \left( \boldsymbol{I}_N - \eta \bar{\mathcal{K}}_{\boldsymbol{\varsigma},\boldsymbol{z}} \right) \cdot [f_{\theta_t}(\boldsymbol{x}_i) - f^*(\boldsymbol{x}_i)]_N
\end{aligned} \tag{21}
$$

where $\bar{\mathcal{K}}_{\boldsymbol{\varsigma},\boldsymbol{z}} = \mathcal{K}_{\boldsymbol{\varsigma},\boldsymbol{z}}/N$. Consequently, using the relationship of adjacent models as shown in Equation 21, we derive

$$
[f_{\theta_t}(\boldsymbol{x}_i) - f^*(\boldsymbol{x}_i)]_N = \left( \boldsymbol{I}_N - \eta \bar{\mathcal{K}}_{\boldsymbol{\varsigma},\boldsymbol{z}} \right)^t \cdot [f_{\theta_0}(\boldsymbol{x}_i) - f^*(\boldsymbol{x}_i)]_N , \tag{22}
$$

that is,

$$
\begin{aligned}
[f_{\theta_t}(\boldsymbol{x}_i)]_N &= [f^*(\boldsymbol{x}_i)]_N + \left( \boldsymbol{I}_N - \eta \bar{\mathcal{K}}_{\boldsymbol{\varsigma},\boldsymbol{z}} \right)^t \cdot [f_{\theta_0}(\boldsymbol{x}_i)]_N - \left( \boldsymbol{I}_N - \eta \bar{\mathcal{K}}_{\boldsymbol{\varsigma},\boldsymbol{z}} \right)^t [f^*(\boldsymbol{x}_i)]_N \\
&= \left( \boldsymbol{I}_N - \left( \boldsymbol{I}_N - \eta \bar{\mathcal{K}}_{\boldsymbol{\varsigma},\boldsymbol{z}} \right)^t \right) [f^*(\boldsymbol{x}_i)]_N + \left( \boldsymbol{I}_N - \eta \bar{\mathcal{K}}_{\boldsymbol{\varsigma},\boldsymbol{z}} \right)^t [f_{\theta_0}(\boldsymbol{x}_i)]_N ,
\end{aligned} \tag{23}
$$

which is precisely Equation 7.

Given the symmetric and positive definite nature of $\bar{\mathcal{K}}_{\boldsymbol{\varsigma},\boldsymbol{z}}$, it can be orthogonally diagonalized as $\bar{\mathcal{K}}_{\boldsymbol{\varsigma},\boldsymbol{z}} = \boldsymbol{V} \boldsymbol{\Lambda} \boldsymbol{V}^\top$ according to the spectral theorem (Hall, 2013). Here, $\boldsymbol{V} = [\boldsymbol{v}_1, \cdots, \boldsymbol{v}_N]$ is a matrix of column vectors $\boldsymbol{v}_i$, which are the eigenvectors corresponding to the eigenvalues $\lambda_i$, and $\boldsymbol{\Lambda} = \mathrm{diag}(\lambda_1, \cdots, \lambda_N)$ is a diagonal matrix with ordered entries ($\lambda_1 \geq \cdots \geq \lambda_N$). Because $\boldsymbol{I}_N = \boldsymbol{V}\boldsymbol{V}^\top$, we have

$$
\begin{aligned}
\boldsymbol{I}_N - \eta \bar{\mathcal{K}}_{\boldsymbol{\varsigma},\boldsymbol{z}} &= \boldsymbol{V} \left( \boldsymbol{I}_N - \eta \boldsymbol{\Lambda} \right) \boldsymbol{V}^\top \\
\therefore \left( \boldsymbol{I}_N - \eta \bar{\mathcal{K}}_{\boldsymbol{\varsigma},\boldsymbol{z}} \right)^t &= \boldsymbol{V} \left( \boldsymbol{I}_N - \eta \boldsymbol{\Lambda} \right)^t \boldsymbol{V}^\top \\
&= \boldsymbol{V} \begin{pmatrix} (1 - \eta\lambda_1)^t & \cdots & 0 \\ \vdots & \ddots & \vdots \\ 0 & \cdots & (1 - \eta\lambda_N)^t \end{pmatrix} \boldsymbol{V}^\top.
\end{aligned} \tag{24}
$$

Since $\eta$ is a small positive scalar, we conclude that $1 - \eta\lambda_i < 1$ for all $i \in \mathbb{N}_N$. Therefore, we obtain

$$
\lim_{t \to \infty} \begin{pmatrix} (1 - \eta\lambda_1)^t & \cdots & 0 \\ \vdots & \ddots & \vdots \\ 0 & \cdots & (1 - \eta\lambda_N)^t \end{pmatrix} \to \boldsymbol{0}_{N \times N}. \tag{25}
$$

Given the asymptotic behavior of the matrix $(\boldsymbol{I}_N - \eta\boldsymbol{\Lambda})^t$, Equation 23 indicates that as $t \to \infty$,

$$\lim_{t\to\infty} f_{\theta_t} = f^*, \tag{26}$$

which shows the convergence of training within the function space.

## A.3 IMPACT OF HIERARCHICAL LAYER STRUCTURES

To explore the influence of hierarchical layer structures in neural networks while employing ZO optimization, we examine a two-layer linear neural network of width $\omega$ as a specific example, defined by

$$f(\boldsymbol{x};\theta) = \left\langle \theta^{(1)}_{n\times\omega} \cdot \theta^{(2)}_{\omega\times 1}, \boldsymbol{x} \right\rangle,$$

where $\theta := \{\theta^{(1)}, \theta^{(2)}\}$ with a total size of $d = n(\omega + 1)$, and $\theta^{(i)}$ denotes the parameters of the $i$-th layer.

From $(**)$ in Eq. 2 and Eq. 5, it follows that the finite difference $(f(\cdot;\theta_t + \epsilon\boldsymbol{z}) - f(\cdot;\theta_t - \epsilon\boldsymbol{z}))/2\epsilon$ – that is, the estimation of the rate magnitude of change of $f(\boldsymbol{x};\theta)$ with respect to $\theta$ – holds significant importance in ZO optimization. This prompts us to closely examine this term in order to understand the behavior of hierarchical layer structures.

**Proposition 3.** *In a two-layer linear neural network $f(\boldsymbol{x};\theta)$, estimating the rate magnitude of change of $f(\boldsymbol{x};\theta)$ with respect to $\theta$ as per Eq. 3 can be equated to summing the estimates alternately conducted for each layer while keeping the other layers fixed,*

$$\frac{f(\theta_t + \epsilon\boldsymbol{z}) - f(\theta_t - \epsilon\boldsymbol{z})}{2\epsilon} = \frac{f\left(\theta^{(1)}_t + \epsilon\boldsymbol{z}^{(1)}\right) - f\left(\theta^{(1)}_t - \epsilon\boldsymbol{z}^{(1)}\right)}{2\epsilon} + \frac{f\left(\theta^{(2)}_t + \epsilon\boldsymbol{z}^{(2)}\right) - f\left(\theta^{(2)}_t - \epsilon\boldsymbol{z}^{(2)}\right)}{2\epsilon}, \tag{27}$$

*where we omit the argument $\boldsymbol{x}$ and fixed parameters, and $\boldsymbol{z}^{(i)}$ denotes a random direction vector of the same dimensions as $\theta^{(i)}_t$.*

*Proof.* See Proof of Proposition 3 in Appendix B.4. ∎

This implies that for ZO optimization, we can estimate the rate of change either across all parameters simultaneously or layer by layer. The layer-by-layer approach requires less memory but increases training time. The underlying reason for this equivalence lies in the independence of parameters during gradient computation, where the order of layers is also irrelevant. Interestingly, this finding aligns with the positive semi-definite Hermitian linear operator in (Xu et al., 2023) and hierarchical decomposition in (Hu et al., 2020). In contrast, FO methods necessitate propagating gradients backward from later to earlier layers.

However, it is worth noting that this term can be intuitively understood as the square root of NZK. This implies that while it can be decomposed, NZK itself is not a linear function of this term, hence NZK cannot be regarded as the sum of decomposed kernels.

## A.4 DERIVATION FOR LINEARIZED NEURAL NETWORKS

For linearized neural networks (LNNs), $f^{\text{lin}}(\theta_t)$ is defined as

$$f^{\text{lin}}(\theta_t) := f(\theta_0) + \left\langle \frac{f(\theta_0 + \epsilon\boldsymbol{u}) - f(\theta_0 - \epsilon\boldsymbol{u})}{2\epsilon}\boldsymbol{u}, \theta_t - \theta_0 \right\rangle, \tag{28}$$

where the rate of change of $f^{\text{lin}}(\theta_t)$ with respect to $\theta_t$ can be approximated using Equation 3 as

$$
\frac{f^{\text{lin}}(\theta_t + \epsilon\boldsymbol{\zeta}) - f^{\text{lin}}(\theta_t - \epsilon\boldsymbol{\zeta})}{2\epsilon}\boldsymbol{\zeta}
$$

$$
= \frac{f(\theta_0) + \left\langle \frac{f(\theta_0+\epsilon\boldsymbol{u})-f(\theta_0-\epsilon\boldsymbol{u})}{2\epsilon}\boldsymbol{u}, \theta_t + \epsilon\boldsymbol{\zeta} - \theta_0 \right\rangle - \left( f(\theta_0) + \left\langle \frac{f(\theta_0+\epsilon\boldsymbol{u})-f(\theta_0-\epsilon\boldsymbol{u})}{2\epsilon}\boldsymbol{u}, \theta_t - \epsilon\boldsymbol{\zeta} - \theta_0 \right\rangle \right)}{2\epsilon}\boldsymbol{\zeta}
$$

$$
= \frac{\left\langle \frac{f(\theta_0+\epsilon\boldsymbol{u})-f(\theta_0-\epsilon\boldsymbol{u})}{2\epsilon}\boldsymbol{u}, 2\epsilon\boldsymbol{\zeta} \right\rangle}{2\epsilon}\boldsymbol{\zeta}
$$

$$
= \left\langle \frac{f(\theta_0 + \epsilon\boldsymbol{u}) - f(\theta_0 - \epsilon\boldsymbol{u})}{2\epsilon}\boldsymbol{u}, \boldsymbol{\zeta} \right\rangle \boldsymbol{\zeta}. \tag{29}
$$

In this scenario, the magnitude represented by $\left\langle \frac{f(\theta_0+\epsilon\boldsymbol{u})-f(\theta_0-\epsilon\boldsymbol{u})}{2\epsilon}\boldsymbol{u}, \boldsymbol{\zeta} \right\rangle$ is not equivalent to $f^{\text{lin}}(\boldsymbol{\zeta})$, as observed in the context of linear models (see Equation 44).

Similar to Equation 15, we can derive

$$
f^{\text{lin}}_{\theta_{t+1}} - f^{\text{lin}}_{\theta_t}
$$

$$
= -\frac{\eta}{N} \left\langle \frac{f^{\text{lin}}(\cdot;\theta_t + \epsilon\boldsymbol{\zeta}) - f^{\text{lin}}(\cdot;\theta_t - \epsilon\boldsymbol{\zeta})}{2\epsilon}\boldsymbol{\zeta}, \right.
$$
$$
\left. \left[ \frac{\mathcal{L}(f^{\text{lin}}(\boldsymbol{x}_i;\theta_t + \epsilon\boldsymbol{z}), y_i) - \mathcal{L}(f^{\text{lin}}(\boldsymbol{x}_i;\theta_t - \epsilon\boldsymbol{z}), y_i)}{f^{\text{lin}}(\boldsymbol{x}_i;\theta_t + \epsilon\boldsymbol{z}) - f^{\text{lin}}(\boldsymbol{x}_i;\theta_t - \epsilon\boldsymbol{z})} \right]^T_N \left[ \frac{f^{\text{lin}}(\boldsymbol{x}_i;\theta_t + \epsilon\boldsymbol{z}) - f^{\text{lin}}(\boldsymbol{x}_i;\theta_t - \epsilon\boldsymbol{z})}{2\epsilon} \cdot \boldsymbol{z} \right]_N \right\rangle
$$

$$
= -\frac{\eta}{N} \left[ \frac{\mathcal{L}(f^{\text{lin}}(\boldsymbol{x}_i;\theta_t + \epsilon\boldsymbol{z}), y_i) - \mathcal{L}(f^{\text{lin}}(\boldsymbol{x}_i;\theta_t - \epsilon\boldsymbol{z}), y_i)}{f^{\text{lin}}(\boldsymbol{x}_i;\theta_t + \epsilon\boldsymbol{z}) - f^{\text{lin}}(\boldsymbol{x}_i;\theta_t - \epsilon\boldsymbol{z})} \right]^T_N
$$
$$
\left\langle \frac{f^{\text{lin}}(\cdot;\theta_t + \epsilon\boldsymbol{\zeta}) - f^{\text{lin}}(\cdot;\theta_t - \epsilon\boldsymbol{\zeta})}{2\epsilon}\boldsymbol{\zeta}, \left[ \frac{f^{\text{lin}}(\boldsymbol{x}_i;\theta_t + \epsilon\boldsymbol{z}) - f^{\text{lin}}(\boldsymbol{x}_i;\theta_t - \epsilon\boldsymbol{z})}{2\epsilon} \cdot \boldsymbol{z} \right]_N \right\rangle
$$

$$
= -\frac{\eta}{N} \left[ \frac{\mathcal{L}(f^{\text{lin}}(\boldsymbol{x}_i;\theta_t + \epsilon\boldsymbol{z}), y_i) - \mathcal{L}(f^{\text{lin}}(\boldsymbol{x}_i;\theta_t - \epsilon\boldsymbol{z}), y_i)}{f^{\text{lin}}(\boldsymbol{x}_i;\theta_t + \epsilon\boldsymbol{z}) - f^{\text{lin}}(\boldsymbol{x}_i;\theta_t - \epsilon\boldsymbol{z})} \right]^T_N
$$
$$
\left\langle \left\langle \frac{f(\cdot;\theta_0 + \epsilon\boldsymbol{u}) - f(\cdot;\theta_0 - \epsilon\boldsymbol{u})}{2\epsilon}\boldsymbol{u}, \boldsymbol{\zeta} \right\rangle \boldsymbol{\zeta}, \left[ \left\langle \frac{f(\boldsymbol{x}_i;\theta_0 + \epsilon\boldsymbol{u}) - f(\boldsymbol{x}_i;\theta_0 - \epsilon\boldsymbol{u})}{2\epsilon}\boldsymbol{u}, \boldsymbol{z} \right\rangle \boldsymbol{z} \right]_N \right\rangle
$$

$$
= -\frac{\eta}{N} \left[ \frac{\mathcal{L}(f^{\text{lin}}(\boldsymbol{x}_i;\theta_t + \epsilon\boldsymbol{z}), y_i) - \mathcal{L}(f^{\text{lin}}(\boldsymbol{x}_i;\theta_t - \epsilon\boldsymbol{z}), y_i)}{f^{\text{lin}}(\boldsymbol{x}_i;\theta_t + \epsilon\boldsymbol{z}) - f^{\text{lin}}(\boldsymbol{x}_i;\theta_t - \epsilon\boldsymbol{z})} \right]^T_N
$$
$$
\left[ \underbrace{\left\langle \left\langle \frac{f(\cdot;\theta_0 + \epsilon\boldsymbol{u}) - f(\cdot;\theta_0 - \epsilon\boldsymbol{u})}{2\epsilon}\boldsymbol{u}, \boldsymbol{\zeta} \right\rangle \boldsymbol{\zeta}, \left\langle \frac{f(\boldsymbol{x}_i;\theta_0 + \epsilon\boldsymbol{u}) - f(\boldsymbol{x}_i;\theta_0 - \epsilon\boldsymbol{u})}{2\epsilon}\boldsymbol{u}, \boldsymbol{z} \right\rangle \boldsymbol{z} \right\rangle}_{(\Gamma)} \right]_N, \tag{30}
$$

where $(\Gamma)$ defines the NZK of LNNs, with its entry being

$$
K_{\boldsymbol{u},\boldsymbol{\zeta},\boldsymbol{z}}(\boldsymbol{x}_i, \boldsymbol{x}_j) = \left\langle \left\langle \frac{f(\boldsymbol{x}_i;\theta_0 + \epsilon\boldsymbol{u}) - f(\boldsymbol{x}_i;\theta_0 - \epsilon\boldsymbol{u})}{2\epsilon}\boldsymbol{u}, \boldsymbol{\zeta} \right\rangle \boldsymbol{\zeta}, \left\langle \frac{f(\boldsymbol{x}_j;\theta_0 + \epsilon\boldsymbol{u}) - f(\boldsymbol{x}_j;\theta_0 - \epsilon\boldsymbol{u})}{2\epsilon}\boldsymbol{u}, \boldsymbol{z} \right\rangle \boldsymbol{z} \right\rangle
$$
$$
= \left\langle \frac{f(\boldsymbol{x}_i;\theta_0 + \epsilon\boldsymbol{u}) - f(\boldsymbol{x}_i;\theta_0 - \epsilon\boldsymbol{u})}{2\epsilon}\boldsymbol{u}, \boldsymbol{\zeta} \right\rangle \left\langle \frac{f(\boldsymbol{x}_j;\theta_0 + \epsilon\boldsymbol{u}) - f(\boldsymbol{x}_j;\theta_0 - \epsilon\boldsymbol{u})}{2\epsilon}\boldsymbol{u}, \boldsymbol{z} \right\rangle \langle \boldsymbol{\zeta}, \boldsymbol{z} \rangle \tag{31}
$$

Following the steps outlined in the proof of Theorem 1 in Appendix B.2 and employing the trace operation method, we can establish that

$$
\mathbb{E}_{\boldsymbol{u},\boldsymbol{\zeta},\boldsymbol{z}}K_{\boldsymbol{u},\boldsymbol{\zeta},\boldsymbol{z}}(\boldsymbol{x}_i, \boldsymbol{x}_j) = \left\langle \underbrace{\mathbb{E}_{\boldsymbol{u}}\frac{f(\boldsymbol{x}_i;\theta_0 + \epsilon\boldsymbol{u}) - f(\boldsymbol{x}_i;\theta_0 - \epsilon\boldsymbol{u})}{2\epsilon}\boldsymbol{u}, \mathbb{E}_{\boldsymbol{u}}\frac{f(\boldsymbol{x}_j;\theta_0 + \epsilon\boldsymbol{u}) - f(\boldsymbol{x}_j;\theta_0 - \epsilon\boldsymbol{u})}{2\epsilon}\boldsymbol{u}}_{(\Phi)} \right\rangle. \tag{32}
$$

We observe that $(\Phi)$ can serve as a ZO estimation of $\frac{\partial f(\boldsymbol{x}_i;\theta)}{\partial \theta_0}$. Thus, by defining $\widehat{\frac{\partial f(\boldsymbol{x}_i;\theta)}{\partial \theta_0}} :=$
$\mathbb{E}_{\boldsymbol{u}} \frac{f(\boldsymbol{x}_i;\theta_0+\epsilon\boldsymbol{u})-f(\boldsymbol{x}_i;\theta_0-\epsilon\boldsymbol{u})}{2\epsilon}\boldsymbol{u}$, we obtain

$$\mathbb{E}_{\boldsymbol{u},\boldsymbol{\varsigma},\boldsymbol{z}} K_{\boldsymbol{u},\boldsymbol{\varsigma},\boldsymbol{z}}(\boldsymbol{x}_i,\boldsymbol{x}_j) = \boldsymbol{\mathcal{K}}_{\boldsymbol{u},\boldsymbol{\varsigma},\boldsymbol{z}}(\boldsymbol{x}_i,\boldsymbol{x}_j) = \left\langle \frac{\widehat{\partial f(\boldsymbol{x}_i;\theta)}}{\partial \theta_0}, \frac{\widehat{\partial f(\boldsymbol{x}_j;\theta)}}{\partial \theta_0} \right\rangle. \tag{33}$$

Following the procedure to derive Equation 7 for linear models, we can also derive

$$\left[ f_{\theta_t}^{\text{lin}}(\boldsymbol{x}_i) \right]_N = \left( \boldsymbol{I}_N - \left( \boldsymbol{I}_N - \eta\bar{\boldsymbol{\mathcal{K}}}_{\boldsymbol{u},\boldsymbol{\varsigma},\boldsymbol{z}} \right)^t \right) [f^*(\boldsymbol{x}_i)]_N + \left( \boldsymbol{I}_N - \eta\bar{\boldsymbol{\mathcal{K}}}_{\boldsymbol{u},\boldsymbol{\varsigma},\boldsymbol{z}} \right)^t \left[ f_{\theta_0}^{\text{lin}}(\boldsymbol{x}_i) \right]_N. \tag{34}$$

Since $f_{\theta_0}^{\text{lin}} = f_{\theta_0}$, we have

$$\left[ f_{\theta_t}^{\text{lin}}(\boldsymbol{x}_i) \right]_N = \left( \boldsymbol{I}_N - \left( \boldsymbol{I}_N - \eta\bar{\boldsymbol{\mathcal{K}}}_{\boldsymbol{u},\boldsymbol{\varsigma},\boldsymbol{z}} \right)^t \right) [f^*(\boldsymbol{x}_i)]_N + \left( \boldsymbol{I}_N - \eta\bar{\boldsymbol{\mathcal{K}}}_{\boldsymbol{u},\boldsymbol{\varsigma},\boldsymbol{z}} \right)^t \left[ f_{\theta_0}(\boldsymbol{x}_i) \right]_N, \tag{35}$$

which is exactly Equation 12.

### A.5 THE PROPERTY OF HOMOGENEOUS ACTIVATION

In this paper, we explore the evolution of neural networks through ZO optimization, as detailed in Equation 12, assuming sufficiently large width (Jacot et al., 2018; Arora et al., 2019). However, practical neural networks are generally of finite width. To reconcile the disparity between theoretical infinite-width neural networks and real-world finite-width ones, the property of homogeneous activation (Du et al., 2018; Lyu & Li, 2019) can be pivotal. This property links to linear models through nonlinear gating (Tian, 2023).

**Property 4.** *(Du et al., 2018) The activation $\varphi(\cdot)$ is homogeneous if it satisfies $\varphi(x) = \varphi'(x)x$.*

We can verify that this characteristic holds for functions such as ReLU $\varphi(x) = \max\{0,x\}$, Leaky ReLU $\varphi(x) = \max\{\alpha x, x\}, \alpha \in (0,1)$, and linear $\varphi(x) = kx$. However, this pertains specifically to FO scenarios. For ZO cases, we directly present the corresponding ZO counterpart as

**Property 5.** *(Zeroth-order Homogeneous) The activation $\varphi(\cdot)$ is zeroth-order homogeneous if it holds $\varphi(x) = \frac{\varphi(x+\epsilon)-\varphi(x-\epsilon)}{2\epsilon}x$, where $\epsilon$ is a sufficiently small positive constant.*

We can also validate that these functions (ReLU, Leaky ReLU and linear) satisfy this property: for $x > 0$, we can always find a sufficiently small positive constant $\epsilon$ such that $x-\epsilon > 0$, and similarly for $x < 0$. Specifically, for ReLU $\varphi(x) = \max\{x,0\} = \mathbb{I}_{x\geq0}x$, where $\mathbb{I}$ denotes an indicator function, it has

$$
\begin{aligned}
\frac{\varphi(x+\epsilon) - \varphi(x-\epsilon)}{2\epsilon}x &= \frac{\mathbb{I}_{x+\epsilon\geq0}(x+\epsilon) - \mathbb{I}_{x-\epsilon\geq0}(x-\epsilon)}{2\epsilon}x \\
&= \frac{\mathbb{I}_{x\geq0}(x+\epsilon) - \mathbb{I}_{x\geq0}(x-\epsilon)}{2\epsilon}x \\
&= \frac{\mathbb{I}_{x\geq0}\left((x+\epsilon) - (x-\epsilon)\right)}{2\epsilon}x \\
&= \frac{\mathbb{I}_{x\geq0}(2\epsilon)}{2\epsilon}x \\
&= \varphi(x).
\end{aligned}
\tag{36}
$$

Similarly, for Leaky ReLU $\varphi(x) = \mathbb{I}_{x\geq0}(1-\alpha)x + \alpha x$ $(0 < \alpha < 1)$, it can be verified that

$$
\begin{aligned}
\frac{\varphi(x+\epsilon) - \varphi(x-\epsilon)}{2\epsilon}x &= \frac{\mathbb{I}_{x\geq0}(1-\alpha)(x+\epsilon) + \alpha(x+\epsilon) - \mathbb{I}_{x\geq0}(1-\alpha)(x-\epsilon) + \alpha(x-\epsilon)}{2\epsilon}x \\
&= \frac{\mathbb{I}_{x\geq0}(1-\alpha)(x+\epsilon-x+\epsilon) + \alpha(x+\epsilon-x+\epsilon)}{2\epsilon}x \\
&= \mathbb{I}_{x\geq0}(1-\alpha)x + \alpha x = \varphi(x).
\end{aligned}
\tag{37}
$$

# B    DETAILED PROOFS

Prior to exploring the stability of the NZK, we introduce a lemma regarding the commutativity (*i.e.*, interchangeability) of expectation and trace operations applied to a random matrix (Tropp, 2012).

**Lemma 6.** *Suppose $M_{d\times d}$ is a $d \times d$ matrix where each entry $m_{i,j}$ is a random variable. Then, the expected value of the trace of $M_{d\times d}$ equals the trace of the expectation of $M_{d\times d}$.*

$$\mathbb{E}[\mathrm{Tr}(M_{d\times d})] = \mathrm{Tr}(\mathbb{E}[M_{d\times d}]). \tag{38}$$

With this lemma,

## B.1    PROOF OF LEMMA 6

The trace of random matrix $M_{d\times d}$ is

$$\mathrm{Tr}(M_{d\times d}) = \sum_{i=1}^{d} m_{i,i}. \tag{39}$$

One can observe that the trace of a random matrix is also a random variables. Therefore, taking expectation on both side of above equation, we have

$$\mathbb{E}[\mathrm{Tr}(M_{d\times d})] = \mathbb{E}\left[\sum_{i=1}^{d} m_{i,i}\right]. \tag{40}$$

It follows from the linearity of the expectation that

$$\mathbb{E}\left[\sum_{i=1}^{d} m_{i,i}\right] = \sum_{i=1}^{d} \mathbb{E}[m_{i,i}]. \tag{41}$$

Note that $\sum_{i=1}^{d} \mathbb{E}[m_{i,i}]$ can be expressed as the trace of a $d \times d$ matrix with diagonal entries to be $\mathbb{E}[m_{i,i}], i \in \mathbb{N}_d$. The diagonal entries of the expectation of $M_{d\times d}$ are precisely $\mathbb{E}[m_{i,i}], i \in \mathbb{N}_d$, that is

$$\mathrm{Tr}(\mathbb{E}[M_{d\times d}]) = \sum_{i=1}^{d} \mathbb{E}[m_{i,i}]. \tag{42}$$

Hence, combining Eq. 40-42, we have

$$\mathbb{E}[\mathrm{Tr}(M_{d\times d})] = \mathrm{Tr}(\mathbb{E}[M_{d\times d}]). \tag{43}$$

■

## B.2    PROOF OF THEOREM 1

It follows from Equation 5, *i.e.*, Equation 16 that NZK is predominantly determined by the difference term. In the context of linear models, we can simplify this expression with a vector $\dagger \in \mathbb{R}^d$ as

$$\begin{aligned} f(x;\theta_t + \epsilon\dagger) - f(x;\theta_t - \epsilon\dagger) &= \langle\theta_t + \epsilon\dagger, x\rangle - \langle\theta_t - \epsilon\dagger, x\rangle \\ &= \langle 2\epsilon\dagger, x\rangle \\ &= 2\epsilon f(x;\dagger). \end{aligned} \tag{44}$$

The substitution of $\theta_t$ with $\dagger$ aligns with the approach taken by Nesterov & Spokoiny, 2017, in which $\theta_t$ is replaced by $z$.

By defining inputs $x_i$ and $x_j$, we are able to scrutinize each component of $K_{\zeta,z}$ as

$$\begin{aligned} K_{\zeta,z}(x_i, x_j) &= \langle\langle\zeta, x_i\rangle\zeta, \langle z, x_j\rangle z\rangle \\ &= \langle\zeta, x_i\rangle\langle z, x_j\rangle\langle\zeta, z\rangle \\ &= \langle\zeta, x_i\rangle\langle x_j, z\rangle\langle z, \zeta\rangle \\ &= \zeta^\top x_i x_j^\top z z^\top \zeta. \end{aligned} \tag{45}$$

It is trivial to observe that this is a scalar, so it equals the result after performing the trace operation, indicating

$$
\begin{aligned}
K_{\boldsymbol{\zeta},\boldsymbol{z}}(\boldsymbol{x}_i,\boldsymbol{x}_j) &= \mathrm{Tr}\left(K_{\boldsymbol{\zeta},\boldsymbol{z}}(\boldsymbol{x}_i,\boldsymbol{x}_j)\right) \\
&= \mathrm{Tr}\left(\boldsymbol{\zeta}^\top \boldsymbol{x}_i \boldsymbol{x}_j^\top \boldsymbol{z}\boldsymbol{z}^\top \boldsymbol{\zeta}\right).
\end{aligned}
\tag{46}
$$

Due to the cyclic property of trace operation, we have

$$
K_{\boldsymbol{\zeta},\boldsymbol{z}}(\boldsymbol{x}_i,\boldsymbol{x}_j) = \mathrm{Tr}\left(\boldsymbol{\zeta}\boldsymbol{\zeta}^\top \boldsymbol{x}_i \boldsymbol{x}_j^\top \boldsymbol{z}\boldsymbol{z}^\top\right).
\tag{47}
$$

To control the effects of randomness in estimation and uncover a statistical property of NZK, we compute the expected $K_{\boldsymbol{\zeta},\boldsymbol{z}}(\boldsymbol{x}_i,\boldsymbol{x}_j)$, which involves

$$
\mathbb{E}_{\boldsymbol{\zeta},\boldsymbol{z}}\left[K_{\boldsymbol{\zeta},\boldsymbol{z}}(\boldsymbol{x}_i,\boldsymbol{x}_j)\right] = \mathbb{E}_{\boldsymbol{\zeta},\boldsymbol{z}}\left[\mathrm{Tr}\left(\boldsymbol{\zeta}\boldsymbol{\zeta}^\top \boldsymbol{x}_i \boldsymbol{x}_j^\top \boldsymbol{z}\boldsymbol{z}^\top\right)\right].
\tag{48}
$$

Using Lemma 6, we obtain

$$
\mathbb{E}_{\boldsymbol{\zeta},\boldsymbol{z}}\left[\mathrm{Tr}\left(\boldsymbol{\zeta}\boldsymbol{\zeta}^\top \boldsymbol{x}_i \boldsymbol{x}_j^\top \boldsymbol{z}\boldsymbol{z}^\top\right)\right] = \mathrm{Tr}\left(\mathbb{E}_{\boldsymbol{\zeta},\boldsymbol{z}}\left[\boldsymbol{\zeta}\boldsymbol{\zeta}^\top \boldsymbol{x}_i \boldsymbol{x}_j^\top \boldsymbol{z}\boldsymbol{z}^\top\right]\right).
\tag{49}
$$

Since $\boldsymbol{\zeta} \sim \mathcal{N}(\mu_{\boldsymbol{\zeta}}\mathbf{1}, \sigma_{\boldsymbol{\zeta}}^2 \boldsymbol{I}_d)$ is independent of $\boldsymbol{z} \sim \mathcal{N}(\mu_{\boldsymbol{z}}\mathbf{1}, \sigma_{\boldsymbol{z}}^2 \boldsymbol{I}_d)$, we have

$$
\mathrm{Tr}\left(\mathbb{E}_{\boldsymbol{\zeta},\boldsymbol{z}}\left[\boldsymbol{\zeta}\boldsymbol{\zeta}^\top \boldsymbol{x}_i \boldsymbol{x}_j^\top \boldsymbol{z}\boldsymbol{z}^\top\right]\right) = \mathrm{Tr}\left(\mathbb{E}_{\boldsymbol{\zeta}}\left[\boldsymbol{\zeta}\boldsymbol{\zeta}^\top\right] \boldsymbol{x}_i \boldsymbol{x}_j^\top \mathbb{E}_{\boldsymbol{z}}\left[\boldsymbol{z}\boldsymbol{z}^\top\right]\right).
\tag{50}
$$

Since $\mathbb{E}\left[r^2\right] = \mathbb{V}(r) + \mathbb{E}^2\left[r\right]$ holds for the random variable $r$, the same relationship holds for entries of $\mathbb{E}_{\boldsymbol{z}}\left[\boldsymbol{z}\boldsymbol{z}^\top\right]$, where

$$
\begin{aligned}
\mathbb{E}_{\boldsymbol{z}}\left[\boldsymbol{z}\boldsymbol{z}^\top\right] &= \mathbb{E}_{\boldsymbol{z}}\left[\begin{pmatrix} z_1^2 & \cdots & z_1 z_d \\ \vdots & \ddots & \vdots \\ z_d z_1 & \cdots & z_d^2 \end{pmatrix}\right] \\
&= \begin{pmatrix} \mathbb{E}_{\boldsymbol{z}}\left[z_1^2\right] & \cdots & \mathbb{E}_{\boldsymbol{z}}\left[z_1 z_d\right] \\ \vdots & \ddots & \vdots \\ \mathbb{E}_{\boldsymbol{z}}\left[z_d z_1\right] & \cdots & \mathbb{E}_{\boldsymbol{z}}\left[z_d^2\right] \end{pmatrix} \\
&= \begin{pmatrix} \sigma_{\boldsymbol{z}}^2 + \mu_{\boldsymbol{z}}^2 & \cdots & \mu_{\boldsymbol{z}}^2 \\ \vdots & \ddots & \vdots \\ \mu_{\boldsymbol{z}}^2 & \cdots & \sigma_{\boldsymbol{z}}^2 + \mu_{\boldsymbol{z}}^2 \end{pmatrix} \\
&= \sigma_{\boldsymbol{z}}^2 \boldsymbol{I}_n + \mu_{\boldsymbol{z}}^2 \mathbf{1}_{n\times n}.
\end{aligned}
\tag{51}
$$

Similarly, for $\mathbb{E}_{\boldsymbol{\zeta}}\left[\boldsymbol{\zeta}\boldsymbol{\zeta}^\top\right]$, we can derive that $\mathbb{E}_{\boldsymbol{\zeta}}\left[\boldsymbol{\zeta}\boldsymbol{\zeta}^\top\right] = \sigma_{\boldsymbol{\zeta}}^2 \boldsymbol{I}_n + \mu_{\boldsymbol{\zeta}}^2 \mathbf{1}_{n\times n}$.

Thus, given the linearity of the Trace operation, we have

$$
\begin{aligned}
&\mathrm{Tr}\left(\mathbb{E}_{\boldsymbol{\zeta}}\left[\boldsymbol{\zeta}\boldsymbol{\zeta}^\top\right] \boldsymbol{x}_i \boldsymbol{x}_j^\top \mathbb{E}_{\boldsymbol{z}}\left[\boldsymbol{z}\boldsymbol{z}^\top\right]\right) \\
=~ & \mathrm{Tr}\left(\left(\sigma_{\boldsymbol{\zeta}}^2 \boldsymbol{I}_n + \mu_{\boldsymbol{\zeta}}^2 \mathbf{1}_{n\times n}\right) \boldsymbol{x}_i \boldsymbol{x}_j^\top \left(\sigma_{\boldsymbol{z}}^2 \boldsymbol{I}_n + \mu_{\boldsymbol{z}}^2 \mathbf{1}_{n\times n}\right)\right) \\
=~ & \mathrm{Tr}\left(\sigma_{\boldsymbol{\zeta}}^2 \sigma_{\boldsymbol{z}}^2 \boldsymbol{x}_i \boldsymbol{x}_j^\top\right) + \mathrm{Tr}\left(\sigma_{\boldsymbol{\zeta}}^2 \mu_{\boldsymbol{z}}^2 \boldsymbol{x}_i \boldsymbol{x}_j^\top \mathbf{1}_{n\times n}\right) + \mathrm{Tr}\left(\mu_{\boldsymbol{\zeta}}^2 \sigma_{\boldsymbol{z}}^2 \mathbf{1}_{n\times n} \boldsymbol{x}_i \boldsymbol{x}_j^\top\right) + \mathrm{Tr}\left(\mu_{\boldsymbol{\zeta}}^2 \mu_{\boldsymbol{z}}^2 \mathbf{1}_{n\times n} \boldsymbol{x}_i \boldsymbol{x}_j^\top \mathbf{1}_{n\times n}\right) \\
=~ & \sigma_{\boldsymbol{\zeta}}^2 \sigma_{\boldsymbol{z}}^2 \langle \boldsymbol{x}_i, \boldsymbol{x}_j\rangle + \sigma_{\boldsymbol{\zeta}}^2 \mu_{\boldsymbol{z}}^2 \langle \boldsymbol{x}_i, \mathbf{1}\rangle\langle \mathbf{1}, \boldsymbol{x}_j\rangle + \mu_{\boldsymbol{\zeta}}^2 \sigma_{\boldsymbol{z}}^2 \langle \boldsymbol{x}_i, \mathbf{1}\rangle\langle \mathbf{1}, \boldsymbol{x}_j\rangle + \mu_{\boldsymbol{\zeta}}^2 \mu_{\boldsymbol{z}}^2 d\langle \boldsymbol{x}_i, \mathbf{1}\rangle\langle \mathbf{1}, \boldsymbol{x}_j\rangle.
\end{aligned}
\tag{52}
$$

Therefore, by combining the equations above, we have

$$
\mathbb{E}_{\boldsymbol{\zeta},\boldsymbol{z}}K_{\boldsymbol{\zeta},\boldsymbol{z}}(\boldsymbol{x}_i,\boldsymbol{x}_j) = \sigma_{\boldsymbol{\zeta}}^2 \sigma_{\boldsymbol{z}}^2 \langle \boldsymbol{x}_i, \boldsymbol{x}_j\rangle + \sigma_{\boldsymbol{\zeta}}^2 \mu_{\boldsymbol{z}}^2 \langle \boldsymbol{x}_i, \mathbf{1}\rangle\langle \mathbf{1}, \boldsymbol{x}_j\rangle + \mu_{\boldsymbol{\zeta}}^2 \sigma_{\boldsymbol{z}}^2 \langle \boldsymbol{x}_i, \mathbf{1}\rangle\langle \mathbf{1}, \boldsymbol{x}_j\rangle + \mu_{\boldsymbol{\zeta}}^2 \mu_{\boldsymbol{z}}^2 d\langle \boldsymbol{x}_i, \mathbf{1}\rangle\langle \mathbf{1}, \boldsymbol{x}_j\rangle,
\tag{53}
$$

which concludes the proof.

∎

Assuming that the components of $\boldsymbol{z}$ and $\boldsymbol{\zeta}$ have zero mean and unit variance, denoted as $\mu_{\boldsymbol{z}} = \mu_{\boldsymbol{\zeta}} = 0$ and $\sigma_{\boldsymbol{z}}^2 = \sigma_{\boldsymbol{\zeta}}^2 = 1$, Equation 53 simplifies to $\mathbb{E}_{\boldsymbol{\zeta},\boldsymbol{z}}K_{\boldsymbol{\zeta},\boldsymbol{z}}(\boldsymbol{x}_i,\boldsymbol{x}_j) = \langle \boldsymbol{x}_i, \boldsymbol{x}_j\rangle$

Meanwhile, the expectation can be realized by exploiting batch sampling (Duchi et al., 2015; Liu et al., 2020c),

$$
\mathcal{G}_{\text{batch}} = \frac{1}{B}\sum_{j=1}^{B}\left(\frac{1}{N}\sum_{i=1}^{N}\frac{\mathcal{L}(f(\boldsymbol{x}_i;\theta_t+\epsilon\boldsymbol{z}_j),y_i) - \mathcal{L}(f(\boldsymbol{x}_i;\theta_t-\epsilon\boldsymbol{z}_j),y_i)}{2\epsilon}\boldsymbol{z}_j\right)
\tag{54}
$$

with a sufficiently large batch size $B$, as per the law of large numbers.

### B.3   PROOF OF COROLLARY 2

When generating $\boldsymbol{z}$ and setting $\boldsymbol{\zeta}$ to be equal to $\boldsymbol{z}$, *i.e.*, $\boldsymbol{\zeta} = \boldsymbol{z}$, for each entry, we have

$$
\begin{aligned}
\mathbb{E}_{\boldsymbol{\zeta},\boldsymbol{z}}\left[K_{\boldsymbol{\zeta},\boldsymbol{z}}(\boldsymbol{x}_i,\boldsymbol{x}_j)\right] &= \mathbb{E}_{\boldsymbol{\zeta},\boldsymbol{z}}\left[\boldsymbol{\zeta}^\top \boldsymbol{x}_i \boldsymbol{x}_j^\top \boldsymbol{z}\boldsymbol{z}^\top \boldsymbol{\zeta}\right] \\
&= \mathbb{E}_{\boldsymbol{z}}\left[\boldsymbol{z}^\top \boldsymbol{x}_i \boldsymbol{x}_j^\top \boldsymbol{z}\boldsymbol{z}^\top \boldsymbol{z}\right].
\end{aligned} \tag{55}
$$

Since $\boldsymbol{z}^\top \boldsymbol{x}_i \boldsymbol{x}_j^\top \boldsymbol{z}\boldsymbol{z}^\top \boldsymbol{z}$ results in a scalar value, upon performing the trace operation, we obtain

$$
\mathbb{E}_{\boldsymbol{z}}\left[\boldsymbol{z}^\top \boldsymbol{x}_i \boldsymbol{x}_j^\top \boldsymbol{z}\boldsymbol{z}^\top \boldsymbol{z}\right] = \mathbb{E}_{\boldsymbol{z}}\left[\mathrm{Tr}\left(\boldsymbol{z}^\top \boldsymbol{x}_i \boldsymbol{x}_j^\top \boldsymbol{z}\boldsymbol{z}^\top \boldsymbol{z}\right)\right]. \tag{56}
$$

Due to the cyclic property of trace operation, we have

$$
\mathrm{Tr}\left(\boldsymbol{z}^\top \boldsymbol{x}_i \boldsymbol{x}_j^\top \boldsymbol{z}\boldsymbol{z}^\top \boldsymbol{z}\right) = \mathrm{Tr}\left(\boldsymbol{z}\boldsymbol{z}^\top \boldsymbol{z}\boldsymbol{z}^\top \boldsymbol{x}_i \boldsymbol{x}_j^\top\right). \tag{57}
$$

Thus, employing Lemma 6, we obtain

$$
\begin{aligned}
\mathbb{E}_{\boldsymbol{z}}\left[\mathrm{Tr}\left(\boldsymbol{z}\boldsymbol{z}^\top \boldsymbol{z}\boldsymbol{z}^\top \boldsymbol{x}_i \boldsymbol{x}_j^\top\right)\right] &= \mathrm{Tr}\left(\mathbb{E}_{\boldsymbol{z}}\left[\boldsymbol{z}\boldsymbol{z}^\top \boldsymbol{z}\boldsymbol{z}^\top \boldsymbol{x}_i \boldsymbol{x}_j^\top\right]\right) \\
&= \mathrm{Tr}\left(\mathbb{E}_{\boldsymbol{z}}\left[\boldsymbol{z}\boldsymbol{z}^\top \boldsymbol{z}\boldsymbol{z}^\top\right]\boldsymbol{x}_i \boldsymbol{x}_j^\top\right).
\end{aligned} \tag{58}
$$

Regarding $\mathbb{E}_{\boldsymbol{z}}\left[\boldsymbol{z}\boldsymbol{z}^\top \boldsymbol{z}\boldsymbol{z}^\top\right]$, let us examine the specific formulation

$$
\begin{aligned}
\mathbb{E}_{\boldsymbol{z}}\left[\boldsymbol{z}\boldsymbol{z}^\top \boldsymbol{z}\boldsymbol{z}^\top\right] &= \mathbb{E}_{\boldsymbol{z}}\left[\sum_{i=1}^{d} z_i^2 \begin{pmatrix} z_1^2 & \cdots & z_1 z_d \\ \vdots & \ddots & \vdots \\ z_d z_1 & \cdots & z_d^2 \end{pmatrix}\right] \\
&= \begin{pmatrix} \mathbb{E}_{\boldsymbol{z}}\left[z_1^4 + \sum_{i=2}^{d} z_1^2 z_i^2\right] & \cdots & \mathbb{E}_{\boldsymbol{z}}\left[z_1^3 z_d + z_d^3 z_1 + \sum_{i=2}^{d-1} z_1 z_d z_i^2\right] \\ \vdots & \ddots & \vdots \\ \mathbb{E}_{\boldsymbol{z}}\left[z_d z_1^3 + z_1 z_d^3 + \sum_{i=2}^{d-1} z_1 z_d z_i^2\right] & \cdots & \mathbb{E}_{\boldsymbol{z}}\left[z_d^4 + \sum_{i=1}^{d-1} z_d^2 z_i^2\right] \end{pmatrix} \\
&= \begin{pmatrix} \mathbb{E}_{\boldsymbol{z}}\left[z_1^4\right] + \sum_{i=2}^{d} \mathbb{E}_{\boldsymbol{z}}\left[z_1^2\right]\mathbb{E}_{\boldsymbol{z}}\left[z_i^2\right] & \cdots & 0 \\ \vdots & \ddots & \vdots \\ 0 & \cdots & \mathbb{E}_{\boldsymbol{z}}\left[z_d^4\right] + \sum_{i=1}^{d-1} \mathbb{E}_{\boldsymbol{z}}\left[z_d^2\right]\mathbb{E}_{\boldsymbol{z}}\left[z_i^2\right] \end{pmatrix} \\
&= \begin{pmatrix} \mathbb{E}_{\boldsymbol{z}}\left[z_1^4\right] + (d-1)\mathbb{E}_{\boldsymbol{z}}^2\left[z_1^2\right] & \cdots & 0 \\ \vdots & \ddots & \vdots \\ 0 & \cdots & \mathbb{E}_{\boldsymbol{z}}\left[z_d^4\right] + (d-1)\mathbb{E}_{\boldsymbol{z}}^2\left[z_d^2\right] \end{pmatrix},
\end{aligned} \tag{59}
$$

where $z_i$ are independent and identically distributed entries of $\boldsymbol{z}$ with zero mean.

Hence, Equation 59 can be rewritten as

$$
\mathbb{E}_{\boldsymbol{z}}\left[\boldsymbol{z}\boldsymbol{z}^\top \boldsymbol{z}\boldsymbol{z}^\top\right] = \left(\mathbb{E}_{\boldsymbol{z}}\left[z_i^4\right] + (d-1)\mathbb{E}_{\boldsymbol{z}}^2\left[z_i^2\right]\right)\boldsymbol{I}_d = \left(\mathbb{V}z_i^2 + d \cdot \mathbb{E}^2\left[z_i^2\right]\right)\boldsymbol{I}_d, \tag{60}
$$

where $\mathbb{V}$ denotes the variance operation and $i \in \mathbb{N}_d$. Therefore, we have

$$
\begin{aligned}
\mathrm{Tr}\left(\mathbb{E}_{\boldsymbol{z}}\left[\boldsymbol{z}\boldsymbol{z}^\top \boldsymbol{z}\boldsymbol{z}^\top\right]\boldsymbol{x}_i \boldsymbol{x}_j^\top\right) &= \mathrm{Tr}\left(\left(\mathbb{V}z_i^2 + d \cdot \mathbb{E}^2\left[z_i^2\right]\right)\boldsymbol{I}_d \boldsymbol{x}_i \boldsymbol{x}_j^\top\right) \\
&= \left(\mathbb{V}z_i^2 + d \cdot \mathbb{E}^2\left[z_i^2\right]\right)\langle \boldsymbol{x}_i, \boldsymbol{x}_j\rangle,
\end{aligned} \tag{61}
$$

which means

$$
\mathbb{E}_{\boldsymbol{\zeta},\boldsymbol{z}}\left[K_{\boldsymbol{\zeta},\boldsymbol{z}}(\boldsymbol{x}_i,\boldsymbol{x}_j)\right] = \left(\mathbb{V}z_i^2 + d \cdot \mathbb{E}^2\left[z_i^2\right]\right)\langle \boldsymbol{x}_i, \boldsymbol{x}_j\rangle. \tag{62}
$$

If $\boldsymbol{z} \sim \mathcal{N}(\boldsymbol{0}, \sigma_{\boldsymbol{z}}^2 \boldsymbol{I}_d)$, *i.e.*, $z_i \sim \mathcal{N}(0, \sigma_{\boldsymbol{z}}^2), i \in \mathbb{N}_d$, for $\mathbb{E}_{\boldsymbol{z}}\left[z_i^4\right] + (d-1)\mathbb{E}_{\boldsymbol{z}}^2\left[z_i^2\right]$, we obtain

$$
\begin{aligned}
\mathbb{E}_{\boldsymbol{z}}\left[z_i^4\right] + (d-1)\mathbb{E}_{\boldsymbol{z}}^2\left[z_i^2\right] &= \sigma_{\boldsymbol{z}}^4 \mathbb{E}_{\boldsymbol{z}}\left[\left(\frac{z_i}{\sigma_{\boldsymbol{z}}}\right)^4\right] + (d-1)\sigma_{\boldsymbol{z}}^4 \mathbb{E}_{\boldsymbol{z}}^2\left[\left(\frac{z_i}{\sigma_{\boldsymbol{z}}}\right)^2\right] \\
&= \sigma_{\boldsymbol{z}}^4 \left(\mathbb{V}_{\boldsymbol{z}}\left(\frac{z_i}{\sigma_{\boldsymbol{z}}}\right)^2 + \mathbb{E}_{\boldsymbol{z}}^2\left(\frac{z_i}{\sigma_{\boldsymbol{z}}}\right)^2\right) + (d-1)\sigma_{\boldsymbol{z}}^4 \mathbb{E}_{\boldsymbol{z}}^2\left[\left(\frac{z_i}{\sigma_{\boldsymbol{z}}}\right)^2\right] \\
&= \sigma_{\boldsymbol{z}}^4 \mathbb{V}_{\boldsymbol{z}}\left(\frac{z_i}{\sigma_{\boldsymbol{z}}}\right)^2 + d\sigma_{\boldsymbol{z}}^4 \mathbb{E}_{\boldsymbol{z}}^2\left[\left(\frac{z_i}{\sigma_{\boldsymbol{z}}}\right)^2\right] \\
&= 2\sigma_{\boldsymbol{z}}^4 + d\sigma_{\boldsymbol{z}}^4 \\
&= (d+2)\sigma_{\boldsymbol{z}}^4,
\end{aligned} \tag{63}
$$

where $\left(\frac{z_i}{\sigma_z}\right)^2 \sim \chi^2(1)$ with a variance of 2. Hence, Equation 59 can be specified as

$$\mathbb{E}_{\boldsymbol{z}}\left[\boldsymbol{z}\boldsymbol{z}^\top \boldsymbol{z}\boldsymbol{z}^\top\right] = (d+2)\sigma_{\boldsymbol{z}}^4 \boldsymbol{I}_d. \tag{64}$$

Therefore, we have

$$\begin{aligned} \mathrm{Tr}\left(\mathbb{E}_{\boldsymbol{z}}\left[\boldsymbol{z}\boldsymbol{z}^\top \boldsymbol{z}\boldsymbol{z}^\top\right]\boldsymbol{x}_i\boldsymbol{x}_j^\top\right) &= \mathrm{Tr}\left((d+2)\sigma_{\boldsymbol{z}}^4 \boldsymbol{I}_d \boldsymbol{x}_i \boldsymbol{x}_j^\top\right) \\ &= (d+2)\sigma_{\boldsymbol{z}}^4 \langle \boldsymbol{x}_i, \boldsymbol{x}_j \rangle, \end{aligned} \tag{65}$$

which means

$$\mathbb{E}_{\boldsymbol{\zeta},\boldsymbol{z}}\left[K_{\boldsymbol{\zeta},\boldsymbol{z}}(\boldsymbol{x}_i, \boldsymbol{x}_j)\right] = (d+2)\sigma_{\boldsymbol{z}}^4 \langle \boldsymbol{x}_i, \boldsymbol{x}_j \rangle. \tag{66}$$

∎

This demonstrates that the expected NZK remains constant over time. At each step, assuming initial points $\theta_t$ are the same for ZO and FO methods, then ZO optimization will scale the expected NZK by $(d+2)\sigma_{\boldsymbol{z}}^4$. Moreover, the Gaussian distribution considered here can be extended to other distributions, such as the uniform distribution over a unit ball.

### B.4    PROOF OF PROPOSITION 3

Let's consider how Eq. 3 estimates the magnitude of the rate of change of two-layer linear neural networks $f(\boldsymbol{x};\theta)$ with respect to $\theta$.

$$\begin{aligned} \frac{f(\boldsymbol{x};\theta_t + \epsilon\boldsymbol{z}) - f(\boldsymbol{x};\theta_t - \epsilon\boldsymbol{z})}{2\epsilon} &= \frac{\left(\theta_t^{(2)} + \epsilon\boldsymbol{z}^{(2)}\right)\cdot\left(\theta_t^{(1)} + \epsilon\boldsymbol{z}^{(1)}\right)}{2\epsilon}\boldsymbol{x} - \frac{\left(\theta_t^{(2)} - \epsilon\boldsymbol{z}^{(2)}\right)\cdot\left(\theta_t^{(1)} - \epsilon\boldsymbol{z}^{(1)}\right)}{2\epsilon}\boldsymbol{x} \\ &= \frac{\left(\theta_t^{(2)}\theta_t^{(1)} + \epsilon\theta_t^{(2)}\boldsymbol{z}^{(1)} + \epsilon\boldsymbol{z}^{(2)}\theta_t^{(1)} + \epsilon^2\boldsymbol{z}^{(2)}\boldsymbol{z}^{(1)}\right)}{2\epsilon}\boldsymbol{x} \\ &\quad - \frac{\left(\theta_t^{(2)}\theta_t^{(1)} - \epsilon\theta_t^{(2)}\boldsymbol{z}^{(1)} - \epsilon\boldsymbol{z}^{(2)}\theta_t^{(1)} + \epsilon^2\boldsymbol{z}^{(2)}\boldsymbol{z}^{(1)}\right)}{2\epsilon}\boldsymbol{x} \\ &= \underbrace{\theta_t^{(2)}\boldsymbol{z}^{(1)}\boldsymbol{x}}_{\text{①}} + \underbrace{\boldsymbol{z}^{(2)}\theta_t^{(1)}\boldsymbol{x}}_{\text{②}}. \end{aligned} \tag{67}$$

Let's consider what happens if we fix the second layer and then proceed with estimation. We get

$$\begin{aligned} \frac{f\left(\boldsymbol{x};\theta_t^{(1)} + \epsilon\boldsymbol{z}^{(1)}, \theta_t^{(2)}\right) - f\left(\boldsymbol{x};\theta_t^{(1)} - \epsilon\boldsymbol{z}^{(1)}, \theta_t^{(2)}\right)}{2\epsilon} &= \frac{\theta_t^{(2)}\cdot\left(\theta_t^{(1)} + \epsilon\boldsymbol{z}^{(1)}\right)}{2\epsilon}\boldsymbol{x} - \frac{\theta_t^{(2)}\cdot\left(\theta_t^{(1)} - \epsilon\boldsymbol{z}^{(1)}\right)}{2\epsilon}\boldsymbol{x} \\ &= \frac{\theta_t^{(2)}\theta_t^{(1)} + \epsilon\theta_t^{(2)}\boldsymbol{z}^{(1)}}{2\epsilon}\boldsymbol{x} - \frac{\theta_t^{(2)}\theta_t^{(1)} - \epsilon\theta_t^{(2)}\boldsymbol{z}^{(1)}}{2\epsilon}\boldsymbol{x} \\ &= \theta_t^{(2)}\boldsymbol{z}^{(1)}\boldsymbol{x}, \end{aligned} \tag{68}$$

which is equivalent to ①. Likewise, we can derive analogous results by fixing the first layer as

$$\begin{aligned} \frac{f\left(\boldsymbol{x};\theta_t^{(1)}, \theta_t^{(2)} + \epsilon\boldsymbol{z}^{(2)}\right) - f\left(\boldsymbol{x};\theta_t^{(1)}, \theta_t^{(2)} - \epsilon\boldsymbol{z}^{(2)}\right)}{2\epsilon} &= \frac{\left(\theta_t^{(2)} + \epsilon\boldsymbol{z}^{(2)}\right)\cdot\theta_t^{(1)}}{2\epsilon}\boldsymbol{x} - \frac{\left(\theta_t^{(2)} - \epsilon\boldsymbol{z}^{(2)}\right)\cdot\theta_t^{(1)}}{2\epsilon}\boldsymbol{x} \\ &= \frac{\theta_t^{(2)}\theta_t^{(1)} + \epsilon\boldsymbol{z}^{(2)}\theta_t^{(1)}}{2\epsilon}\boldsymbol{x} - \frac{\theta_t^{(2)}\theta_t^{(1)} - \epsilon\boldsymbol{z}^{(2)}\theta_t^{(1)}}{2\epsilon}\boldsymbol{x} \\ &= \boldsymbol{z}^{(2)}\theta_t^{(1)}\boldsymbol{x}, \end{aligned} \tag{69}$$

which corresponds to ②. Therefore, in the context of a two-layer linear neural network $f(\boldsymbol{x};\theta)$, estimating the rate magnitude of change of $f(\boldsymbol{x};\theta)$ with respect to $\theta$ using Equation 3 is interchangeable with summing the estimates obtained alternately for each layer while holding the other layers fixed,

$$\frac{f(\theta_t + \epsilon\boldsymbol{z}) - f(\theta_t - \epsilon\boldsymbol{z})}{2\epsilon} = \frac{f\left(\theta_t^{(1)} + \epsilon\boldsymbol{z}^{(1)}\right) - f\left(\theta_t^{(1)} - \epsilon\boldsymbol{z}^{(1)}\right)}{2\epsilon} + \frac{f\left(\theta_t^{(2)} + \epsilon\boldsymbol{z}^{(2)}\right) - f\left(\theta_t^{(2)} - \epsilon\boldsymbol{z}^{(2)}\right)}{2\epsilon}. \tag{70}$$

∎

## C    Experiment Details

### C.1    Computing infrastructure

All experiments in this paper conducted on the machine with Ubuntu 20.04 system, equiped with AMD Ryzen 3975WX CPU and NVIDIA RTX3090 GPU (24G). The MNIST, CIFAR-10 and tiny imagenet datasets and corresponding pre-processing code used in our experiments can be obtained from Datasets library provided on Hugging Face.

### C.2    Linear models

The Gaussian perturbation $\delta$ in the teacher model $f^*(\boldsymbol{x}; \theta^*) = 7.0 * x_1 + 2.0 * x_2 + \delta$ follows $\mathcal{N}(0, 0.02)$, where $0.02$ controls the relative magnitude of the perturbation with respect to the training points.

We display the final trained models under FO and ZOs with varying variances after 16,000 iterations in Figure 4, while the individual evolution is shown in Figure 5. Figure 6 presents the NZK comparisons for different $\sigma_{\boldsymbol{z}}$ in $d = 2$.

For the case where $\boldsymbol{\zeta}$ and $\boldsymbol{z}$ are identical, we show the evolution of linear models with different $\sigma_{\boldsymbol{z}}$ in Figure 7, and the corresponding loss comparisons are presented in Figure 1 (b).

To investigate how $d$ affects ZO optimization, as described in Equation 8 from Corollary 2, we perform experiments with varying dimensions $d$ for $z$. Specifically, we set $d \in \{10, 30, 50\}$, meaning $\theta \in \mathbb{R}^{10}/\mathbb{R}^{30}/\mathbb{R}^{50}$. The data points have the same dimension and are normalized, so $\boldsymbol{x} \in \mathbb{S}^{10}/\mathbb{S}^{30}/\mathbb{S}^{50}$. We visualize the evolution by projecting the high-dimensional space into 2D, as shown in Figure 8, with the loss plot in Figure 9 and the NZK top view in Figure 10.

We investigate scenarios where $\boldsymbol{\zeta}$ and $\boldsymbol{z}$ are identical, extending beyond Gaussian distributions to include Laplace and Student's t distributions. The loss results are depicted in Figure 11, illustrating that ZO with identical $\boldsymbol{\zeta}$ and $\boldsymbol{z}$ converges faster than the evolution of FO. Interestingly, Gaussian-1, Laplace-0.604, and Student's t-1000 demonstrate identical convergence performance. This alignment arises from our careful specification of distribution parameters to ensure consistency in $\mathbb{V}z_i^2 + d \cdot \mathbb{E}^2\left[z_i^2\right]$ in Equation 8, illustrating the distributional irrelevance highlighted in Corollary 2. For the Laplace distribution, we use $\mu = 0$ and $b = 0.605$, and for the Student's t distribution, we set the parameter $\nu = 1,000$. Figure 12 illustrates the model's evolution, while Figure 13 compares the final trained models across the different distributions. The 2-D visualization of NZK is shown in Figure 14.

For the MNIST dataset classification using a linear model, the loss is shown in Figure 15, which indicates that updating with traditional parametric gradient ZO converges more slowly than FO, and performs worse compared to using kernel gradients ZO. In this task with $d = 24$, Figure 16 displays the 2-D visualization of NZK.

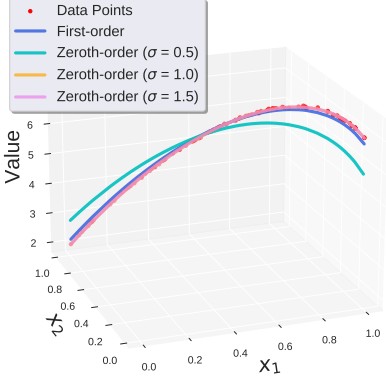

Figure 4: Comparison of the final linear models trained with FO and ZO after 16,000 iterations, with different $\sigma_{\boldsymbol{z}}$. The vectors $\boldsymbol{\zeta}$ and $\boldsymbol{z}$ are sampled independently.

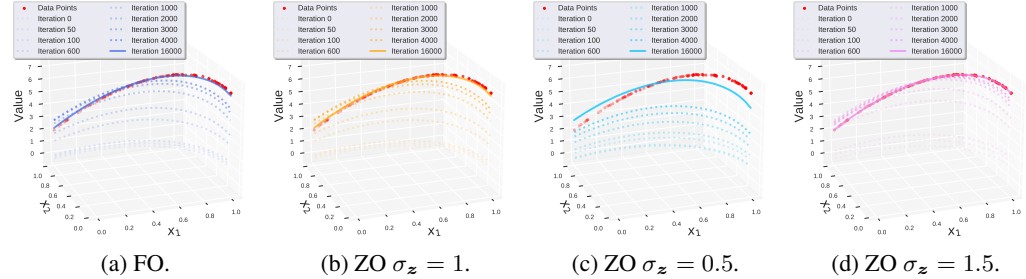

Figure 5: Evolution of linear models under FO and ZO in a 2-D fitting task. The vectors $\zeta$ and $z$ are independently sampled. (a) Evolution under FO. (b-d) Evolution under ZO with (b) $\sigma_z = 1$, (c) $\sigma_z = 0.5$, and (d) $\sigma_z = 1.5$.

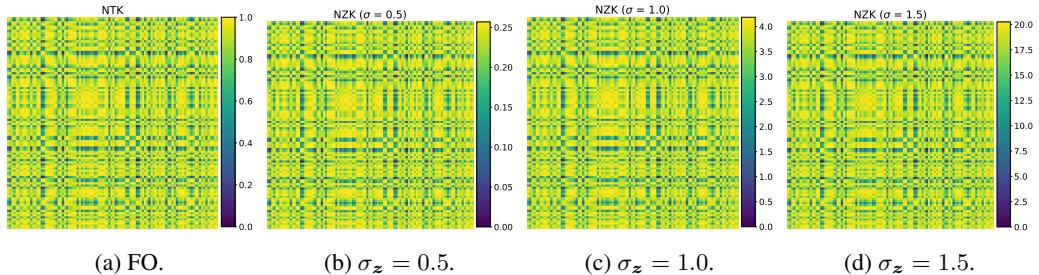

Figure 6: Comparison of NTK and expected NZK with varying $\sigma_z$ for the case of $d = 2$, using identical $\zeta$ and $z$.

### C.3 LINEARIZED NEURAL NETWORKS

We examine a linearized neural network defined as:

$$f^{\text{lin}}(\theta_t) := f(\theta_0) + \left\langle \frac{f(\theta_0 + \epsilon u) - f(\theta_0 - \epsilon u)}{2\epsilon} u, \theta_t - \theta_0 \right\rangle,$$

with $\theta \in \mathbb{R}^{51}$ and $x \in \mathbb{S}^2$. The original nonlinear neural network is a three-layer feed-forward network with ReLU activation applied after the hidden layer.

For the digital classification task using the MNIST dataset, we downsample the images to $8 \times 8$ ones and flatten them into 64-dim vectors as inputs. Therefore, the input layer of the original nonlinear neural network has the dimension of 64. This network also includes two hidden layers with 10 and 5 neurons, respectively, and an output layer of dimension 1, resulting in a total of 711 parameters. Figure 2 (a) illustrates the loss while Figure 17 presents a visual comparison of NTK and NZK. For the classification task on the CIFAR-10 and tiny imagenet dataset, we follow the same setting as for MNIST dataset. The visualizations of NZK corresponding to CIFAR-10 are provided in Figure 18.

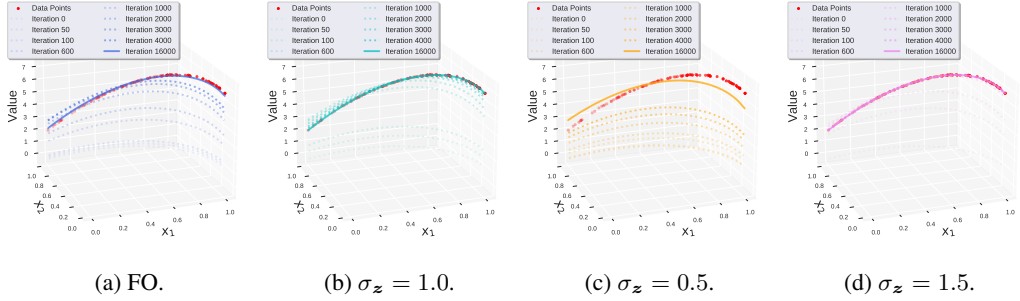

(a) FO.     (b) $\sigma_{\boldsymbol{z}} = 1.0$.     (c) $\sigma_{\boldsymbol{z}} = 0.5$.     (d) $\sigma_{\boldsymbol{z}} = 1.5$.

Figure 7: Comparison of linear model evolution under FO and ZO with different $\sigma_{\boldsymbol{z}}$ for $d = 2$, using identical $\boldsymbol{\zeta}$ and $\boldsymbol{z}$.

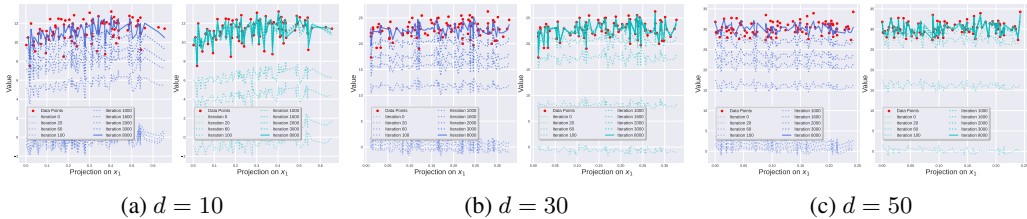

(a) $d = 10$     (b) $d = 30$     (c) $d = 50$

Figure 8: Comparison of linear model evolution under FO and ZO with varying $d$, projected into 2-D space. Identical $\boldsymbol{\zeta}$ and $\boldsymbol{z}$ are used.

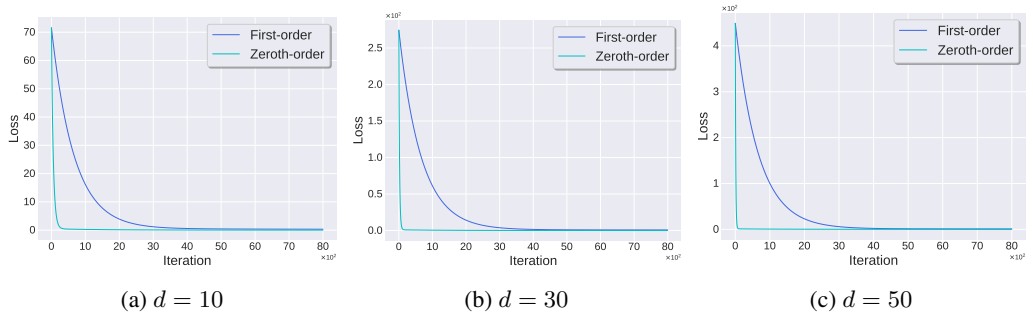

(a) $d = 10$     (b) $d = 30$     (c) $d = 50$

Figure 9: Comparison of the loss for the linear model under FO and ZO, with varying $d$ and identical $\boldsymbol{\zeta}$ and $\boldsymbol{z}$.

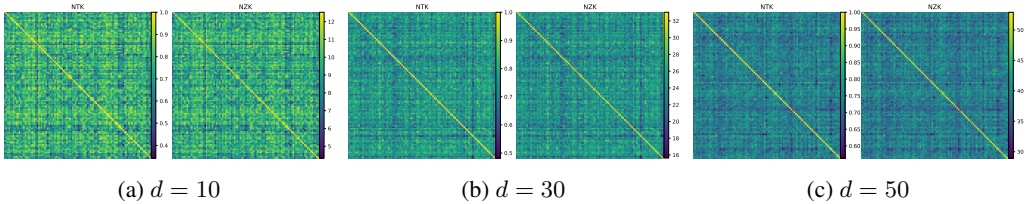

(a) $d = 10$     (b) $d = 30$     (c) $d = 50$

Figure 10: Comparison of NTK and expected NZK for the linear model under FO and ZO, with different values of $d$ and identical $\boldsymbol{\zeta}$ and $\boldsymbol{z}$.

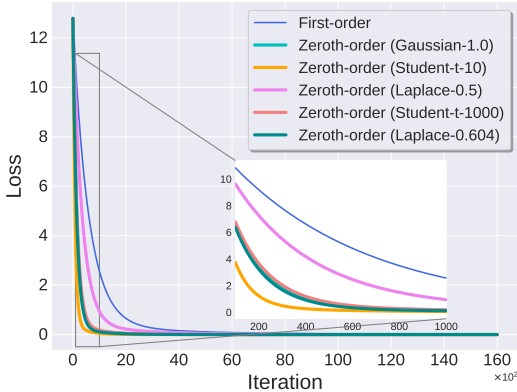

Figure 11: Comparison of losses in 2-D fitting tasks under FO and ZO with different distributions, while keeping $\zeta$ and $z$ identical.

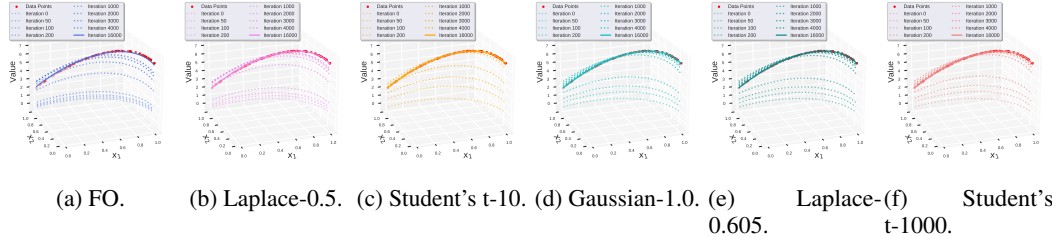

(a) FO.  (b) Laplace-0.5.  (c) Student's t-10.  (d) Gaussian-1.0.  (e) Laplace-0.605.  (f) Student's t-1000.

Figure 12: Comparison of linear model evolution under FO and ZO with different distributions for $d = 2$. Identical $\zeta$ and $z$ are used. The second row shows similar performance.

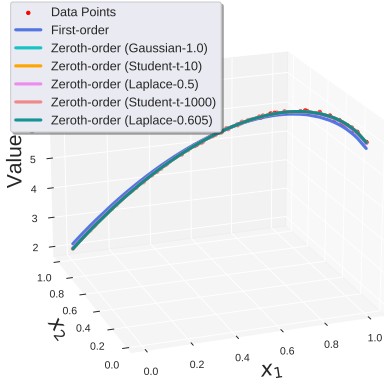

Figure 13: Comparison of the final linear models under FO and ZO after 16,000 iterations with different distributions, for $d = 2$, using identical $\zeta$ and $z$.

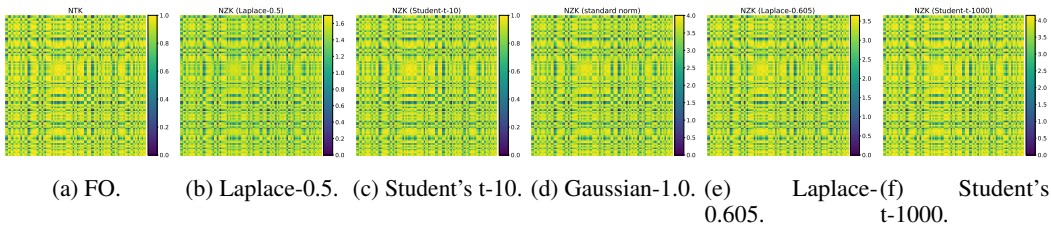

(a) FO.  (b) Laplace-0.5.  (c) Student's t-10.  (d) Gaussian-1.0.  (e) Laplace-0.605.  (f) Student's t-1000.

Figure 14: Comparison of NTK and expected NZK with different distributions for $d = 2$, and identical $\zeta$ and $z$ are taken.

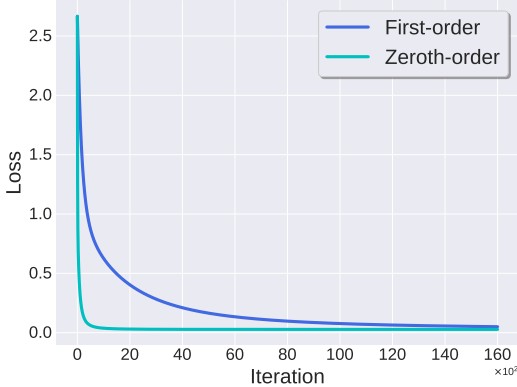

Figure 15: Comparison of losses for the **linear** model under FO and ZO in 24-dimensional classification tasks on MNIST, with identical $\zeta$ and $z$.

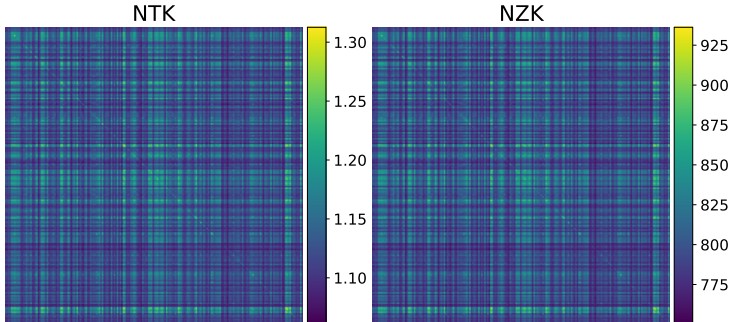

Figure 16: Comparison of NZK for the 24-dimensional **linear** model under FO and ZO for MNIST classification, with identical $\zeta$ and $z$.

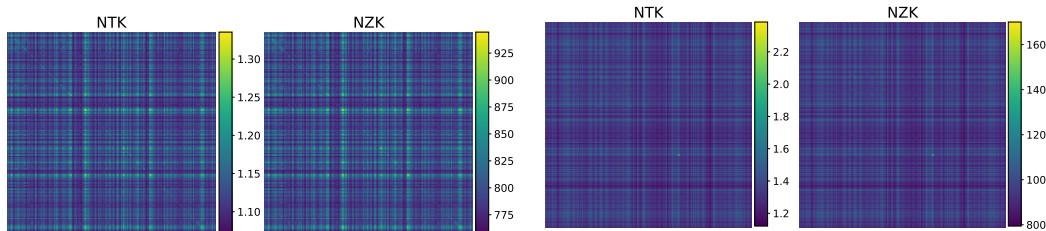

Figure 17: Comparison of NZK for **linearized** neural networks in MNIST classification using FO and ZO, with identical $\zeta$ and $z$.

Figure 18: Comparison of NZK for **linearized** neural networks in CIFAR-10 classification using FO and ZO, with identical $\zeta$ and $z$.

