# OpenReview forum: "Model Evolution Under Zeroth-Order Optimization: A Neural Tangent Kernel Perspective"
_ICLR.cc/2026/Workshop/Sci4DL — Sci4DL 2026_

### Official Review · Reviewer_PYTo · 2026-02-21

**Fit:** 1
**Significance:** 1
**Confidence:** 2

**Summary:**

This paper studies the function-space dynamics of zeroth-order optimization on linear models (or equivalently, on linearized neural networks).  They work out an expression for the expected NTK and they show that the evolution in expectation of the predictions is governed by this expected NTK.  They experimentally compare ZO vs. first-order optimization on linear models and linearized neural nets.

**Strengths:**

The paper addresses a question that has not been studied extensively before.  Moreover, zero-th order optimization is potentially of interest  for its memory efficiency in fine-tuning LLMs

**Suggestions:**

* I don't understand why one would want to use (or analyze) zeroth-order optimization where the noise $z$ has nonzero mean or isotropic  covariance with $\sigma^2$ other than 1, which is a key claimed contribution of the submission.  The reason why Eq (1) works is that up to the finite-difference estimate of the directional derivative, we have $E[\mathcal{G}_t] = E[z z^\top] \nabla \mathcal L$, which equals $\nabla \mathcal L$ if $z \sim N(0, I)$.  If we give $z$ a non-zero mean, as is discussed in the paper, then we no longer have an unbiased estimate of the gradient, and I don't think it'll work.  Further, if $z$ has a zero mean but some non-unit covariance $\sigma^2 I$, then we have an unbiased estimate of $\sigma^2 \nabla \mathcal L$, which is completely equivalent using unit-variance noise while simply rescaling the learning rate.  The submission claims that increasing $\sigma^2$ helps "accelerate ZO optimization", but rescaling the learning rate would have the exact same effect.
* I don't understand the details of the experiment shown in Figure 2(b), but I find it difficult to believe that zero-th order (cyan) is so much better than first order.
* I don't understand the setting we seem to be in, where we need to do a ZO estimate of the model's predictions on the training set while also doing ZO optimization.  (One of the paper's main claimed results is that it is better to use the same noise for both ZO estimates.). Could you please give an example of a real-life situation that exemplifies this setting?
* I believe it is already well-known that if you do SGD (i.e you follow an unbiased estimate of the gradient, as in ZO optimizaton) on a linear model, then the dynamics in expectation are equivalent to that of GD.  This just follows from stepping in the direction of an unbiased estimate of the gradient.

---

### Official Review · Reviewer_xYJi · 2026-02-24

**Fit:** 3
**Significance:** 3
**Confidence:** 2

**Summary:**

This paper provides a kernel picture for zeroth-order optimization, giving a characterization of the NTK-like “Neural Zeroth-order Kernel, NZK.” On linear and linearized networks, the expected NZK is proven to recover the NTK, allowing the NZK to be used to describe a network’s optimization dynamics. The empirical NZK qualitatively reproduces the NTK of linearized networks even on image datasets.

**Strengths:**

The NZK allows for linearized network optimization dynamics to be solved under zeroth-order optimization, in expectation. The motivation behind characterizing the NZK is clear, with the results being presented in a logical manner. The NZK is shown to reproduce the well-studied NTK, opening a lot of doors theoretically. The results appear to be widely applicable to zeroth-order optimization overall.

**Suggestions:**

Zeroth-order optimization is established to be useful for problems where gradients are ill-defined, yet this is never explored empirically. Even without any updates to the theory section, it would be useful to show that the proven theory extends to the use cases of zeroth-order optimization. Variance of randomly sampled vectors for zeroth-order optimization also isn’t discussed, despite being a huge open question of zeroth-order optimization; with "only one z per iteration" (line ~83), it appears plausible that the theoretical results being true in expectation will start failing to account for networks trained on high-dimensional data. Perhaps Figure 3 provides a counterpoint given training on TinyImageNet, but some proper mention of the variance (not) affecting results would be helpful.

Q: On line ~115, it's stated "Eq. 7 simplifies to $\mathcal{K}_{\zeta,z}(x_i, x_j) = ⟨x_i, x_j⟩$", with Eq. 10 yielding a similar ($\mathcal{K} \sim ⟨x_i, x_j⟩$) result when $z = \zeta$; does this imply that zeroth-order optimization on linearized networks is performing linear regression, irrespective of the network's functional form?

---

### Official Review · Reviewer_o3bU · 2026-02-27

**Fit:** 3
**Significance:** 2
**Confidence:** 2

**Summary:**

The paper provides a function space intuition for the zeroth-order (ZO) optimization. It defines a Neural Zeroth-order Kernel (NZK) built from finite-difference directional derivatives under random perturbations, and rewrites ZO output dynamics in a form analogous to kernel gradient descent. For linear models under squared loss, the expected NZK is shown to be constant and yields a closed-form evolution equation; the analysis is later extended to linearized neural networks too. A notable observation is that using the same random vector for both the finite-difference perturbation and the directional derivative in the kernel definition yields a favorable kernel scaling, suggesting potential acceleration. Experiments provide empirical support within these regimes. The theory is clean in its intended setting, but the connection to practical ZO use cases remains weak.

**Strengths:**

- relevant topic: understanding ZO training dynamics is important given growing use in LLM fine-tuning and black-box settings.
- clear core idea: recasting ZO updates in function space is, to my knowledge, a novel and coherent perspective.
- clean results for linear models: the expectation calculation and closed-form dynamics are technically solid with a direct parallel to the FO-NTK case.
- main text overall readable, with only minor presentation issues.

**Suggestions:**

- Single-direction estimation and the learning rate: The gradient estimator uses a single random direction per iteration. Single-direction ZO estimates have variance scaling with dimension, meaning that in practice one either needs multiple directions or careful step-size tuning to make reliable progress. This matters theoretically as for the linearized-network extension, the NZK constancy picture is tied to a regime where the Jacobian stays approximately fixed, which is more plausible when parameters do not move far. High-variance gradient estimates push against this, either through larger steps or through noisier traversal of parameter space. The paper does not discuss this tension. This concern applies only to the linearized model results; for the linear case, it is obviously fine.

- The acceleration claim: When $\zeta = z$, the authors derive $(d+2)$ scaling of the expected kernel and claim this can accelerate convergence. The argument is suggestive, but the link between this expected kernel scaling and practical convergence speed (in terms of variance, step size, and mainly stability) is not established beyond small-scale linearized experiments. It is not clear this holds in practical nonlinear training, and this should be stated more carefully.

- Practical relevance and positioning: It would strengthen the paper to discuss how the NZK perspective could concretely inform modern ZO applications such as LLM fine-tuning; for instance, around perturbation scale or step-size choice. A clearer statement of what NZK adds beyond existing parameter-space ZO convergence theory would also help position the contribution.


- Minor comments: i) Eq.1 appears to have a missing closing parenthesis, ii) odd ordering of figures (Fig 1. comes after Fig 3.).

---

### Meta-Review · Area_Chair_HJZm · 2026-02-28

**Recommendation:** Accept

**Metareview:**

I'm sympathetic to Reviewer PYTo's concerns here: the results do seem to be more of a trivial consequence of the mathematical setup than the exposition suggests. It's not too surprising that the average NZK is constant if the model's linear --- seems kind of inevitable? --- the speedup in the empirics seems too good to be true, and and phrases like "a fascinating discovery of accelerated performance" seem overblown. The memory savings of 0th-order optimization also seem minor in comparison to the cost of getting totally walloped by the curse of dimensionality; how are you getting around that?

Nonetheless, I'm accepting this because I think pushing known analysis to new optimizers --- and especially *dumber* optimizers like 0th-order optimization, as opposed to "smarter" optimizers like Adam --- is a useful direction. In the future, I'd like to see more of a scientific, empirically-grounded approach here rather than a mathematical one.

---

### Decision · Program_Chairs · 2026-03-02

Accept